# Conformal Prediction with Temporal Quantile Adjustments

**Zhen Lin**[1]     **Shubhendu Trivedi**[2*]     **Jimeng Sun**[1,3]

[1] Department of Computer Science, University of Illinois at Urbana-Champaign
[2] Massachusetts Institute of Technology
[3] Carle Illinois College of Medicine, University of Illinois at Urbana-Champaign
{zhenlin4,jimeng}@illinois.edu   shubhendu@csail.mit.edu

## Abstract

We develop Temporal Quantile Adjustment (TQA), a general method to construct efficient and valid prediction intervals (PIs) for regression on cross-sectional time series data. Such data is common in many domains, including econometrics and healthcare. A canonical example in healthcare is predicting patient outcomes using physiological time-series data, where a population of patients composes a cross-section. Reliable PI estimators in this setting must address two distinct notions of coverage: *cross-sectional* coverage across a cross-sectional slice, and *longitudinal* coverage along the temporal dimension for each time series. Recent works have explored adapting Conformal Prediction (CP) to obtain PIs in the time series context. However, none handles both notions of coverage simultaneously. CP methods typically query a pre-specified quantile from the distribution of *non-conformity scores* on a calibration set. TQA adjusts the quantile to query in CP at each time $t$, accounting for both cross-sectional and longitudinal coverage in a theoretically-grounded manner. The post-hoc nature of TQA facilitates its use as a general wrapper around any time series regression model. We validate TQA's performance through extensive experimentation: TQA generally obtains efficient PIs and improves longitudinal coverage while preserving cross-sectional coverage. Our code is available at https://github.com/zlin7/TQA.

## 1   Introduction

The impressive predictive performance of modern "black-box" machine learning methods has started to make them critical ingredients in various high-stakes decision-making pipelines. It is thus increasingly important to quantify the predictive uncertainty of such models reliably and efficiently, which remains a fundamental challenge. Conformal Prediction (CP), pioneered by Vovk et al. [53], is a powerful framework for quantifying uncertainty under mild assumptions. The model-agnostic and distribution-free nature of CP makes it particularly suitable for large neural network models, and has started to attract the attention of the deep learning community [2, 3, 6, 9, 12, 13, 33, 59]. The primary assumption in most current CP methods is that of data exchangeability. For instance, only assuming exchangeability of the calibration and test data, one can construct $1 - \alpha$ valid prediction intervals by simply querying the corresponding quantile of nonconformity scores on the calibration set. Recent works have started exploring the adaptation of CP in settings that go beyond the usual exchangeability assumption [4, 17, 21, 38, 42, 43, 44, 50, 57], and to more complex data such as time series [17, 48, 56, 58].

We study the adaptation of CP to the cross-sectional time series regression setting. More formally, suppose our data comprises of $N$ time series, denoted $\{\mathbf{S}_i\}_{i=1}^N$, with each $\mathbf{S}_i$ sampled from an

---

*During the initiation and pursuance of this research, the author's primary affiliation was MIT.

36th Conference on Neural Information Processing Systems (NeurIPS 2022).

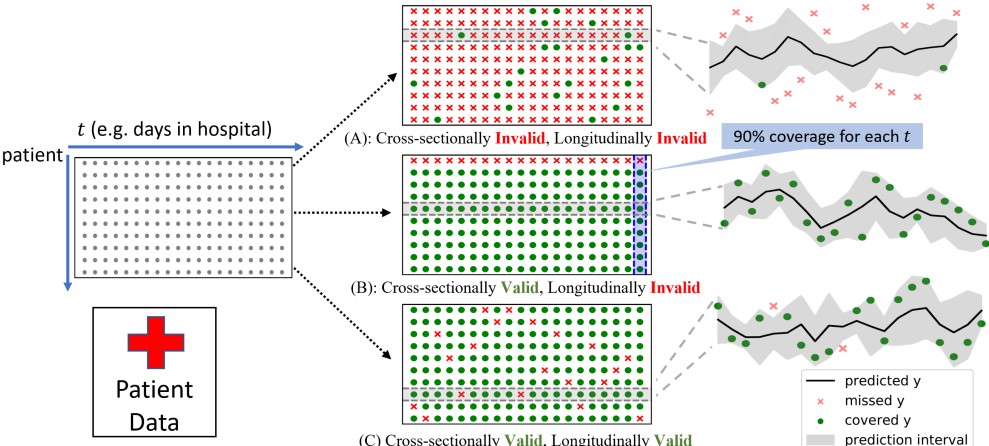

Figure 1: PI estimators with different cross-sectional or longitudinal properties/validities. ✗ denotes the ground-truth $y$ that is outside of the PI. (A) features PIs that are not valid in either sense: $Y$ is never covered. (B) is cross-sectionally valid: for any $t$, 90% of the $Y_{\cdot,t}$ are covered. It is longitudinally *invalid*, as 10% of the TS receive no coverage at all. (C) exhibits both cross-sectional and longitudinal validity, which is ideal.

arbitrary distribution $\mathcal{P}_S$. Further, each time series $\mathbf{S}_i$ is a sequence of temporally-dependent random variables $[Z_{i,1}\ldots,Z_{i,t},\ldots,Z_{i,T}]$, with $Z_{i,t} = (X_{i,t}, Y_{i,t})$ consisting of covariates $X_{\cdot,\cdot} \in \mathbb{R}^d$ and the response $Y_{\cdot,\cdot} \in \mathbb{R}$. Given data until time $t$ ($\{Z_{N+1,j}\}_{j=1}^{t}$) and $X_{N+1,t+1}$ for a new time series $\mathbf{S}_{N+1}$, the time series regression problem entails predicting the response $Y_{N+1,t+1}$ at (an unknown) time $t + 1$. We are interested in quantifying the uncertainty of each prediction by constructing *valid* prediction intervals (PI). That is, given a confidence level $1 - \alpha$, we are interested in constructing a prediction interval, $\hat{C}_{\alpha,N+1,t+1}$, that will cover $Y_{N+1,t+1}$ with probability of at least $1 - \alpha$.

A crucial requirement in cross-sectional time series regression is to distinguish two notions of validity, *longitudinal* and *cross-sectional*, to ensure reliable performance. Longitudinal validity is concerned with validity along the temporal axis for each time series. On the other hand, cross-sectional validity is concerned with validity *across the populational cross-section* of the time series data. Figure 1 illustrates both notions of validity and a standard real-world occurrence of such a problem setting. Recently, several research groups have explored adapting CP to the time series setting. Some of such works [17, 58, 56] focus only on longitudinal validity, which is extremely difficult without strong distributional assumptions [4]. Furthermore, such methods cannot leverage rich information inherent in the cross-section. The only work which addresses cross-sectional validity is to due to Stankevičiūtė et al. [48], but it ignores the temporal dependence. Accounting for both notions of coverage is critical to obtain reliable performance.

To remedy the above situation, we propose Temporal Quantile Adjustment (TQA) for CP in the cross-sectional time series regression setting. TQA is the *first method* that can account for both cross-sectional and longitudinal validity simultaneously. Although TQA can be used as a wrapper around any time series regression model, we focus on neural networks, as our main inspiration comes from complicated time series regression problems in healthcare[2]. Neural networks are particularly suited for such tasks, which can involve modeling the evolution of heterogeneous entities such as diagnostic and drug codes, patient and physician embeddings and regressing over a target of interest. Taking inspiration from [17], TQA adjusts the quantile to query at each time step in a theoretically-grounded manner. Based on the nature of quantile adjustment, we also propose two variants of TQA, which further shed light on the generality of our method. The ability of TQA to handle both cross-sectional and longitudinal validity is borne out in extensive experimentation, where it significantly outperforms competing methods.

## 2 Related Work

Our work falls squarely within the Conformal Prediction (CP) framework. The original formulation of CP was in a purely transductive setting [46, 47, 53], and was computationally inefficient. More

---

[2]We include results for other models in the Appendix.

efficient variants dubbed as Inductive Conformal Prediction (ICP) [39, 40, 52] were proposed soon after and were more broadly popularized by followup works in Statistics [29, 31, 32]. A similar idea, often referred to as Split Conformal [30], is now more or less used interchangeably with ICP, and is fundamental to our paper. The model-agnostic and distribution-free nature of split conformal makes it suitable for large black-box models, and thus it has seen adoption in deep learning-based pipelines (e.g. [2, 11, 26, 33, 36, 13, 48]).

One of the mainstays of the CP framework is the assumption of exchangeability between calibration and test data. Extensions of CP "beyond exchangeability" have attracted relatively little attention until recently. A key publication in the area is [50] which used the notion of weighted exchangeability to handle covariate shift. Several notable works that address various aspects of the covariate shift problem include [21, 42, 57, 44]. More recent work [4, 17] handles gradual distribution drifts. [4] additionally proposes an extension of CP when the data-points cannot be treated in a symmetric manner. Extensions of the split conformal method to a broad class of dependent processes such as stationary $\beta$-mixing processes was proposed by [38]. These works provide some general methodological insights to model temporal dependence in our case, but are otherwise not directly related. In particular, the online adaptive method of [17], served as a major source of inspiration for the development of a variant of TQA (dubbed TQA-E). The most related works have a focus on generating valid intervals in time series regression [17, 58, 4, 56]. These works end up ignoring cross-sectional aspects, which is understandable given the tasks they study. On the other extreme is [48], which focuses only on cross-sectional coverage, ignoring the temporal dimension in constructing PIs.

Various other techniques, outside the ambit of CP, have also been extended to quantify uncertainty in time series forecasting as well. For instance, approximate Bayesian methods [8, 54, 37, 35, 25, 15, 28] are quite popular for uncertainty quantification, and have been extended to RNNs [14, 7]. Finally, one may also use the idea of directly predicting the quantiles (as opposed to the point estimate) in regression tasks [49, 27], and applying it to time series forecasting [55, 16]. However, such methods usually require changing the base model and typically do not come with coverage guarantees.

## 3 Preliminaries

This section builds foundation for our exposition of TQA. We begin by expounding further on longitudinal and cross-sectional validity, followed by presenting the exchangeability assumption. Finally, we discuss the use of split conformal prediction to construct (cross-sectionally) valid PIs.

### 3.1 Cross-sectional Validity vs Longitudinal Validity

**Cross-sectional validity** is the more common type of validity encountered in CP, being the only type of validity in non-time-series settings. More formally:

**Definition 1.** *Prediction interval $\hat{C}_{\cdot,\cdot}$ is $1 - \alpha$ cross-sectionally valid if, for any t,*

$$\mathbb{P}_{\mathbf{S}_{N+1}}\{Y_{N+1,t} \in \hat{C}_{N+1,t}\} \geq 1 - \alpha. \tag{1}$$

$\mathbb{P}_{\mathbf{S}_{N+1}}$ means the probability is taken over the randomness of $\mathbf{S}_{N+1}$. If $\hat{C}_{N+1,t}$ is random (e.g. depends on $\{\mathbf{S}_i\}_{i=1}^N$) then the probability is taken over the randomness of $\hat{C}_{N+1,t}$ as well. Note that if we consider the case where every time series only consists of one step ($T = 1$), then we recover the usual definition of marginal validity. Cross-sectional validity translates to high-probability in coverage for a *randomly drawn time series*. As we will see later, cross-sectional validity is easier to achieve, since we can assume *inter*-time-series exchangeability.

**Longitudinal validity**, on the other hand, is concerned with coverage along the temporal axis for a particular TS. We use the following definition:

**Definition 2.** *Prediction interval $\hat{C}_{\cdot,\cdot}$ is $1 - \alpha$ longitudinally valid if for almost every time-series $\mathbf{S}_{N+1} \sim \mathcal{P}_S$ there exists a $T_0$ such that:*

$$t > T_0 \implies \mathbb{P}_{Y_{N+1,t}|\mathbf{S}_{N+1,:t-1}}\{Y_{N+1,t} \in \hat{C}_{N+1,t}\} \geq 1 - \alpha. \tag{2}$$

Here, the qualifier "almost every" means that the set of time series' for which such coverage may fail is of measure zero (under $\mathcal{P}_S$). The threshold $T_0$ allows some "time" for $\hat{C}$ to potentially adapt to the

temporal information in a particular TS. It should be clear that longitudinal validity is harder to attain, because we can no longer marginalize the probability over randomly drawn time series.

## 3.2 Conformal Prediction for Cross-sectionally Valid PI

In this section we explain how to use conformal prediction to construct cross-sectionally valid PIs. We first introduce exchangeability assumption, which is the central assumption in conformal prediction, and slightly weaker than the standard i.i.d assumption. More formally:

**Definition 3.** *(**Exchangeability** [53]) A sequence of random variables, $Z_1, Z_2, \ldots, Z_n \in \mathcal{Z}$ are exchangeable if for any permutation $\pi : \{1, 2, \ldots, n\} \to \{1, 2, \ldots, n\}$, and every measurable set $E \subseteq \mathcal{Z}^n$, we have*

$$\mathbb{P}\{(Z_1, Z_2, \ldots, Z_n) \in E\} = \mathbb{P}\{(Z_{\pi(1)}, Z_{\pi(2)}, \ldots, Z_{\pi(n)}) \in E\} \tag{3}$$

Definition 3 can naturally be extended to a sequence of randomly drawn time series:

**Definition 4.** *(**Exchangeable Time Series**) Given time series $\mathbf{S}_1, \mathbf{S}_2, \ldots, \mathbf{S}_n$ where $\mathbf{S}_i = [Z_{i,1}, \ldots, Z_{i,T}, \ldots]$, denote $Z_{i,\{t_j\}_{j=1}^m}$ as the concatenated random variable of $(Z_{i,t_1}, \ldots, Z_{i,t_m})$. Time series $\mathbf{S}_1, \mathbf{S}_2, \ldots, \mathbf{S}_n$ are exchangeable if, for any finitely many $t_1 < \cdots < t_m$, the random variables $Z_{1,\{t_j\}_{j=1}^m}, \ldots, Z_{n,\{t_j\}_{j=1}^m}$ are exchangeable.*

As a concrete example, suppose we *randomly* pick 100 patients from a hospital's EHR database for predicting readmission risk. It is fairly reasonable to assume that these time series are exchangeable, despite the obvious strong temporal dependence *within* each time series. *Throughout this paper, we will assume $\mathbf{S}_1, \ldots, \mathbf{S}_{N+1}$ are exchangeable time series.*

We now explain how to construct cross-sectionally valid PIs. To construct a PI for $Y_{i,t}$, we first split our data $\{\mathbf{S}_i\}_{i=1}^N$ into a *proper training set* and a *calibration set* [41]. The training set is used to train models for the *nonconformity score* function, and the calibration set is used to collect such nonconformity scores (denoted as $V(\cdot)$). For example, one may train a mean estimator $\hat{\mu}$ (e.g. an RNN) on the training set, and use the absolute residual $v_{i,t} \leftarrow |y_{i,t} - \hat{y}_{i,t}|$, where $\hat{y}_{i,t} = \hat{\mu}(X_{i,t}; \mathbf{S}_{i,:t-1})$, as the nonconformity score. Here $\mathbf{S}_{i,:t-1}$ denotes $[Z_{i,1}, \ldots, Z_{i,t-1}]$. The idea behind the split conformal method is that the scoring function (e.g. $\hat{\mu}$) is only fit on the proper training set, implying that nonconformity scores on the calibration set and $v_{N+1,t}$ are also *exchangeable*. Here on, for the sake of simplicity of notation, we will use $\{\mathbf{S}_i\}_{i=1}^N$ to denote the *calibration set only*, assuming all necessary models have already been trained.

Given a nonconformity score $V(\cdot)$ (possibly using some trained model $\hat{\mu}$) and a set of exchangeable time series $\{\mathbf{S}_i\}_{i=1}^{N+1}$, the split conformal method can be used to generate the following $1 - \alpha$ cross-sectionally valid prediction interval:

$$\hat{C}_{N+1,t+1}^{split} = \left\{ y : V_{N+1,t+1}(\hat{y}, y) \leq Q\left(1 - \alpha; \{v_{j,t+1}\}_{j=1}^N \cup \{V_{N+1,t+1}(\hat{y}, y)\}\right) \right\}. \tag{4}$$

Here, $Q(\beta; A)$ means the $\beta$-quantile for the set $A$. As a concrete example, if we let $v_{i,t} = |y_{i,t} - \hat{y}_{i,t}|$, and employ the standard trick that replaces $v_{N+1,t+1}$ with $\infty$ to avoid plugging in (uncountably) many values for $y$ [4], the prediction interval for $Y_{N+1,t+1}$ becomes:

$$\hat{C}_{N+1,t+1}^{split} := [\hat{y} - \hat{v}, \hat{y} + \hat{v}] \text{ where } \hat{v} := Q\left(1 - \alpha; \{|y_{i,t+1} - \hat{y}_{i,t+1}|\}_{i=1}^N \cup \{\infty\}\}\right) \tag{5}$$

Assuming exchangeability, we can easily show the cross-sectional validity of $\hat{C}^{split}$:

**Theorem 3.1.** *([4, 53]) $\hat{C}^{split}$ is $1 - \alpha$ cross-sectionally valid (Def. 1).*

The intuition behind the proof is that the exchangeability of the time series translates to the exchangeability of the nonconformity scores, which means the rank of $v_{N+1,t+1}$ among $\{v_{i,t+1}\}_{i=1}^{N+1}$ follows a uniform distribution. The coverage guarantee in Theorem 3.1 then follows. Note that with a finite calibration set, the $1 - \alpha$-quantile could be ambiguously defined, and in practice one would use $\frac{\lceil (1-\alpha)(N+1) \rceil}{N+1}$ to get a slightly more conservative PI, or "flip a (biased) coin" to choose between $\frac{\lceil (1-\alpha)(N+1) \rceil}{N+1}$ and $\frac{\lfloor (1-\alpha)(N+1) \rfloor}{N+1}$ for a precise $1 - \alpha$ coverage (e.g. the "smoothed" ICP in [53] or the tie-breaking trick in [2]). In Section 4, we assume the precise coverage PI for the ease of discussion.

While cross-sectionally valid, $\hat{C}^{split}$ ignores the temporal dependence in the nonconformity scores completely. We will explain in Section 4 how to adapt to the temporal dependence by "quantile adjustment", thus improving longitudinal coverage as well.

# 4 Temporal Quantile Adjustment (TQA)

In Section 3 we discussed a classical conformal prediction method and also highlighted its inherent limitations in the time series setting. In this section we will formally introduce Temporal Quantile Adjustment (TQA), which queries quantiles differently than in the aforementioned split conformal procedures. We first explicate the goals and motivations of TQA in Section 4.1. Then, in Section 4.2 and 4.3 we propose two principled adjustment methods along with theoretical analyses.

## 4.1 Improving Longitudinal Coverage

Although it is tempting to directly pursue distribution-free finite-sample PI estimator that achieves longitudinal *coverage guarantee* (Def. 2), it is likely too optimistic due to the fact that $[Z_{i,1}, \ldots, Z_{i,T}]$ are not exchangeable and we cannot characterize them meaningfully without imposing (strong) distributional assumptions. For example, in [4], the bound for coverage gap—which captures the loss in coverage compared to what is achievable under exchangeability—essentially becomes 1 as the data becomes time-dependent. In fact, a concurrent work [34] proves that longitudinal validity as specified in Def 2 is impossible to achieve unless trivially, following similar proof techniques for the impossibility of distribution-free finite-sample conditional validity [5, 32].

Nevertheless, longitudinal coverage may still be empirically *improved* if we can adapt to the temporal dependence. From our EHR example, suppose $\hat{\mu}$ has been giving high-error predictions for a patient for 8 out of the past 10 days, we might suspect high error going forward for this patient as well. If this is indeed the case, then naively applying $\hat{C}^{split}$ can only attain low coverage for this patient, no matter how long a history we observe. To address this, we propose to query different quantiles based on the partially observed time series.

From now on, we use $\hat{C}_{a_{i,t},i,t}$ to denote a PI for $Y_{i,t}$ with a pre-specified target coverage of $1 - a_{i,t}$, in order to emphasize dependence on the "quantile to query". We let $a_{i,t} = \alpha - \hat{\delta}_{i,t}$, where $\hat{\delta}_{i,t}$ is the *quantile adjustment*. Classical split conformal PIs (e.g. [48]) entail a special case: $\forall i, \forall t, \hat{\delta}_{i,t} \equiv 0$. We refer to this method as Temporal Quantile Adjustment (TQA).

Now, denote the random variable $R_{i,t}$ as the rank/quantile of $V_{i,t}$ among $\{V_{j,t}\}_{j=1}^{N+1}$:

$$r_{i,t} := Q^{-1}(v_{i,t}; \{v_{j,t}\}_{j=1}^{N+1}) := \frac{|\{j : v_{j,t} < v_{i,t}\}|}{N+1}. \tag{6}$$

For example, if $v_{i,t}$ is the smallest among $\{v_{j,t}\}_{j=1}^{N+1}$, then $r_{i,t} = 0$. Intuitively, we would like to use a more conservative (smaller) $a_{N+1,t}$ when we believe $\mathbf{S}_{N+1}$ as a whole is less "conformal" and $R_{N+1,t}$ is likely high. A crucial observation is that if there is no actual temporal dependence between the nonconformity scores (i.e. we are just adjusting $a_t$ based on some "noise"), then we *do not lose any coverage* as long as the expected adjustment is zero.

**Theorem 4.1.** *If the nonconformity score's rank $(R_{N+1,t})$ is independent of the quantile adjustment $(\hat{\delta}_{N+1,t})$, then $\mathbb{P}_{\mathbf{S}_{N+1}}\{Y_{N+1,t} \in \hat{C}_{a_{N+1,t},N+1,t}\} \geq 1 - \alpha + \mathbb{E}[\hat{\delta}_{N+1,t}]$.*

All proofs are deferred to the Appendix.

**Remark**: The assumption in Theorem 4.1 is *not* that $\mathbf{S}_{N+1}$ itself is not temporally dependent, nor is it the slightly weaker assumption, that of the temporal independence of either $\{R_{\cdot,t}\}_{t=1}^{T}$ or $\{V_{\cdot,t}\}_{t=1}^{T}$. The assumption in Theorem 4.1 only suggests that there is no temporal pattern in the prediction errors that $\hat{\delta}$ can capture. This could happen, for example, when the RNN captures the underlying data generating process fully, but only misses the random noise (aleatoric uncertainty). This could also happen if our quantile adjustments $(\hat{\delta})$ are pure noise.

Unfortunately, although it can be tempting to conclude that $\mathbb{E}[\hat{\delta}_{N+1,t}] = 0$ implies finite-sample cross-sectional validity, this conclusion would be incorrect, because it ignores the dependence between

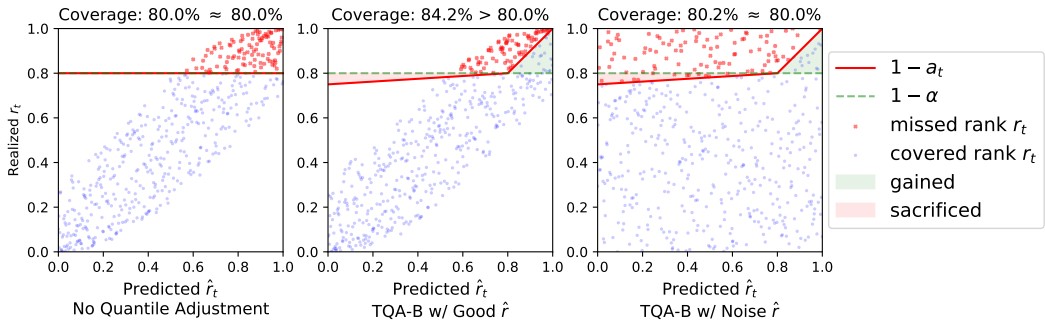

Figure 2: Coverage profiles with hypothetical realized rank $r$ condition on prediction $\hat{r}$, with $\alpha = 0.2$ for readability. ($Y_{i,t} \in \hat{C}_{i,t} \Leftrightarrow r_{i,t} \leq 1 - a_{i,t}$.) As $\hat{r}$ follows a uniform distribution, the proportion of dots below the red line represents the cross-sectional coverage probability. TQA-B generally improves coverage if $\hat{r}$ is correlated with the realized $r$ (middle), and does not lose coverage otherwise (right). "Budgeting" refers to the constraint that sacrificed and gained have equal areas.

$\mathbb{1}\{Y_{N+1,t} \in \hat{C}_{a_{N+1,t}, N+1, t}\}$ and $a_{N+1,t}$. However, we will next discuss how to perform quantile adjustment, and why it typically *improves* coverage.

## 4.2 Quantile Budgeting (TQA-B)

Theorem 4.1 provides an interesting constraint that we should consider while designing $\hat{\delta}$. That is, we should let $\mathbb{E}[\hat{\delta}] = 0$ so as to keep the same coverage when we *cannot* predict the quantiles, but hopefully improving coverage when we can. This suggests a design of $\hat{\delta}$ that we refer to as a type of "budgeting". Although it is possible to directly predict a good $\hat{\delta}_{i,t+1}$, we adopt a more principled two-step approach:

(i) We predict quantile $\hat{r}_{i,t+1}$ which estimates $r_{i,t+1}$.
(ii) We use a pre-defined mapping $g$ to define the quantile adjustment $\hat{\delta}_{i,t+1} \leftarrow g(\hat{r}_{i,t+1}; \alpha)$.

This also permits research to improve each component independently. We introduce one alternative for each step in the Appendix.

**(i) Quantile Prediction**: The quantile prediction $\hat{r}_{i,t+1}$ is estimated by a function of the form $f(\mathbf{S}_{i,:t}; \{\mathbf{S}_{j,:t}\}_{j=1}^{N+1})$. Since $\hat{r}_{i,t+1}$ is supposed to predict $r_{i,t+1}$, *the rank* of the nonconformity score, we impose the constraint that $\hat{r}_{i,t+1}$ should follow a uniform distribution over $\{\frac{j}{N}\}_{j=0}^{N}$. We will focus on a simple rank prediction method stated below:

$$\hat{r}_{i,t+1}^{ms} := Q^{-1}(\bar{\epsilon}_{i,t}; \{\bar{\epsilon}_{j,t}\}_{j=1}^{N+1}) \text{ where } \bar{\epsilon}_{i,t} := \sum_{t'=1}^{t} \frac{|y_{i,t'} - \hat{y}_{i,t'}|}{t} \beta^{(t-t')}. \quad (7)$$

Here, $\bar{\epsilon}_{i,t}$ is the exponentially decayed mean residual of time series $i$ up to time $t$, and we use $\beta = 0.8$. Note that taking the rank with $Q^{-1}$ achieves the uniformity requirement, and $\bar{\epsilon}_{\cdot,t}$ could be replaced by any scoring function that takes into account the temporal information.

**(ii) Budgeting**: Given prediction $\hat{r}_{i,t}$, we propose the following adjustment $\hat{\delta}_{i,t} := g^B(\hat{r}_{i,t}; \alpha)$:

$$g^B(r; \alpha) := \begin{cases} C(r - (1 - \alpha)) & (r < 1 - \alpha) \\ (r - (1 - \alpha)) & (r \geq 1 - \alpha) \end{cases} \text{ where } C = \frac{(2\alpha N - \lfloor \alpha N \rfloor)(\lfloor \alpha N \rfloor + 1)}{\lceil (1 - \alpha)N \rceil((1 - 2\alpha)N + 1 + \lfloor \alpha N \rfloor)}. \quad (8)$$

Denote $a_{N+1,t}^{TQA-B}$ as the quantile to query by TQA-B. Our particular coefficient design ensures that:

**Theorem 4.2.** *Using $\hat{r}_{\cdot,\cdot}^{ms}$ and $g^B$, $\forall t, \mathbb{E}_{\mathbf{S}_{N+1}}[a_{N+1,t}^{TQA-B}] = \alpha$. (Recall $a_{N+1,t}^{TQA-B} = \alpha - g^B(\cdot)$)*

Thus, by Theorem 4.1, TQA-B will not lose coverage if $\hat{r}$ and $r$ are independent. The following theorem provides a worst-case cross-sectional coverage guarantee, regardless of how "bad" $\hat{r}$ is:

**Theorem 4.3.** $\mathbb{P}_{\mathbf{S}_{N+1} \sim \mathcal{P}_S}\{Y_{N+1,t+1} \in \hat{C}_{\alpha,N+1,t+1}^{TQA-B}\} \geq 1 - \alpha - \underbrace{\left(\frac{\alpha + \frac{1}{2N}}{1 - \alpha + \frac{1}{2N}}\right)^2 (1 - \alpha)}_{\text{worst-case loss}}.$

The worst-case loss term is typically small: about $0.012$ for $\alpha = 0.1$ and $N = 100$, although it can also be also be high for a large $\alpha$ like $0.5$. In practice, it is unlikely that $\hat{r}$ is worse than a random guess; the coverage is typically greater than $1 - \alpha$, as we will see in the experiments and illustrated in Figure 2. In the Appendix, we present a more aggressive quantile adjustment function $g$ that provides a weaker guarantee than Theorem 4.3, but empirically performs better.

**Implementation Details** To avoid creating infinitely-wide PIs, we could also let $\hat{\delta} = \lambda g(\hat{r}; \alpha)$ so $a_t$ is bounded away from 0 (In our experiments we choose $\lambda$ such that $a_t \geq 0.01$). Moreover, the specific form of $C$ presented in this section depends on the concrete distribution of $\hat{r}$. For example, $C$ would take a different form if $\hat{r}$ is defined to be uniform over $\{\frac{j}{N+1}\}_{j=1}^{N+1}$ rather than $\{\frac{j}{N}\}_{j=0}^{N}$. Practically, we can simply let $C = \alpha^2(1-\alpha)^{-2}$ regardless of $N$. The additional loss in Theorem 4.3 will become $\alpha^2/(1-\alpha)$, and the change in Theorem 4.2 is negligible for a reasonable value of $N$.

### 4.3 Error-based adjustment (TQA-E)

Another simple quantile adjustment approach is using a heuristic that depends on the past "errors": Define $err_t = \mathbb{1}\{Y_t \notin \hat{C}_{a_t}\}$, and increase $\delta_{t+1}$ (conservative) if we see too many errors in $\{err_{t'}\}_{t'<t+1}$ compared with $\alpha$, and vice versa. Since this approach does not depend on the cross-section, we drop the subscript $\cdot_{N+1}$ for simplicity. We use the following update rule (with $\hat{\delta}_0 = 0$) inspired by [17][3]:

$$\hat{\delta}_{t+1} \leftarrow \begin{cases} \hat{\delta}_t + \gamma(err_t - \alpha) & (\hat{\delta}_t \geq \alpha - 1) \\ (1-\gamma)\hat{\delta}_t & (otherwise) \end{cases}. \tag{9}$$

Note that we do not explicitly impose the restriction that $\alpha - \hat{\delta}_t = a_t \in [0, 1]$, which means the PI could have *infinite width*. However, infinite-wide PI means no error, so $\hat{\delta}_{t+1}$ will decrease and we resume to a finite PI gradually.

As $\hat{\delta}_t$ depends on the entire error history, it is not immediately clear whether TQA-E is still valid with the assumption in Theorem 4.1. Below we state a "no-worse" type theorem for TQA-E:

**Theorem 4.4.** *If we assume the nonconformity score's rank has no temporal dependence, then*

$$\forall t, \mathbb{E}_{\mathbf{S}_{N+1} \sim \mathcal{P}_S}[a_{N+1,t}^{TQA-E}] \leq \alpha. \tag{10}$$

*Thus, $\hat{C}^{TQA-E}$ is finite-sample cross-sectional $1 - \alpha$ valid following Theorem 4.1.*

Finally, an asymptotic longitudinal validity result can also be shown for long time series[4]:

**Theorem 4.5.** *(Asymptotic Longitudinal Coverage) For any time series $\mathbf{S}$, $\lim_{T \to \infty} \frac{\sum_{t=0}^{T-1} err_t}{T} \leq \alpha$.*

**Remarks**: Although Theorem 4.5 seems to suggest some sort of longitudinal validity, it does not contradict the hardness claim in Section 4.1, because TQA-E achieves this via infinitely-wide PIs. We also refer interested readers to [58] as an example of an different heuristic based on errors of a single time series. However, it will require further modifications for finite-sample cross-sectional validity.

## 5 Experiments

In this section, our goal is to to verify the following empirically:

1. TQA maintains cross-sectional coverage and achieves competitive PI efficiency.
2. Ignoring temporal dependence in naive split conformal prediction leads to low longitudinal coverage for some TS.
3. TQA improves longitudinal coverage.

---

[3]Despite the similarity on the surface, [17] has no notion of cross-section.

[4]The update rule as shown in Eq. 9 creates an asymmetry to accommodate for the asymptotic guarantee in Theorem 4.5, which is why the expectation in Theorem 4.4 is not an equality but an inequality. If asymptotic coverage is not a concern (because $T$ is small), one could use $(1-\gamma)\hat{\delta}_t$ when $\alpha - \hat{\delta}_t = a_t < 0$ as well, in which case Theorem 4.4 becomes an equality, as we discuss in the proof for Theorem 4.4 in the Appendix. In practice (as we observe in our experiments), the difference in behavior is negligible.

**Baselines**: We use the following state-of-the-art baselines for PI construction: Conformal forecasting RNN (CFRNN (Split)) [48]), a direct application of split-conformal prediction [53][5]; Quantile RNN (QRNN) [55], which directly predicts the two endpoints (represented by two quantiles) of the PI; RNN with Monte-Carlo Dropout (DP-RNN) [15]; Conformalized Quantile Regression with QRNN (CQRNN) [45], which, as the name suggests is a conformalized version of quantile regression; Locally adaptive split conformal prediction (LASplit) [29], which uses a normalized absolute error as the nonconformity score (we follow the implementation in [45]).

Table 1: Number of TSs in each dataset along with the length.

| Properties | MIMIC | CLAIM | COVID | EEG | GEFCom/GEFCom-R |
|---|---|---|---|---|---|
| # train/cal/test | 192/100/100 | 2393/500/500 | 200/100/80 | 300/100/200 | 1198/200/700 |
| $T$ (length) | 30 | 30 | 30 | 63 | 24 |

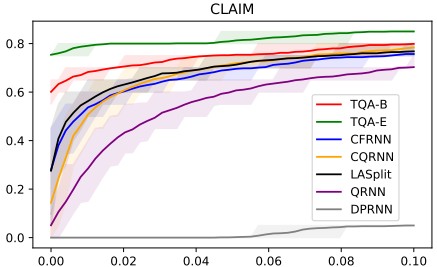

Figure 3: Coverage rate (Y-axis) vs the percentile among all test TS (X-axis, with zero meaning the least-covered TS) for the 10% least-covered TS in CLAIM. The bands denote the center 80% realizations. TQA-E has an natural advantage by using infinitely-wide PIs. However, even TQA-B still significantly improves the longitudinal coverage rate over all baselines.

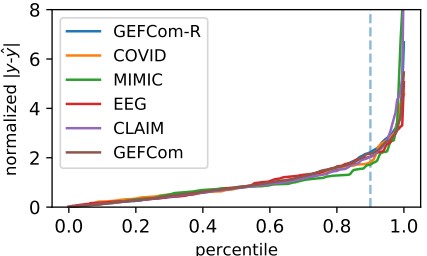

Figure 4: Sorted absolute residuals ($|y - \hat{y}|$) for $t = T - 10$. Each dataset is normalized so the mean of the residuals is 1. To cover extreme values, even if allowed to "sacrifice" some less extreme values, the PI on average is expected to get much wider. It is thus surprising that TQA-B could improve *both* efficiency and the tail coverage.

**Datasets** We test our methods and baselines on the following datasets: Electronic health records data for white blood cell counts (WBCC) prediction (MIMIC [23, 18, 22]), COVID-19 cases prediction (COVID [10]), Electroencephalography trajectory prediction after visual stimuli (EEG [51]), energy load forecasting (GEFCom [20]), and healthcare claim amount prediction (CLAIM) using data from a large American healthcare data provider. Among these, we mostly follow [48] in preparing MIMIC, COVID and EEG. Note that GEFCom is originally a single time series (hourly observations for years). Therefore, we treat each day as a single TS, and perform a strict temporal splitting (test data is preceded by calibration data, which is preceded by the training data), which means *exchangeability is broken*. We also include a GEFCom-R (andom) version that preserves the exchangeability by ignoring the temporal order in data splitting. Table 1 provides a brief summary of the data. Due to space constraints, the details for each dataset are relegated to the Appendix.

**Evaluation Metrics and Experiment Setup** We use RNN as the base point estimator due to its flexibility and for comparison with [48]. We use $\alpha = 0.1$, and a LSTM ([19]) similar to that in [48] (full implementation details in the Appendix). For TQA-E, we use $\gamma = 0.005$ following [17]. For each dataset, we repeat the prediction task 50 times, and report the mean and standard deviation of the average coverage rate, tail coverage rate, and inverse coverage efficiency for the last 20 time-steps. Here, tail coverage rate means the average coverage rate of the least-covered 10% of the time series. A high tail coverage rate thus implies better longitudinal coverage. Inverse coverage efficiency is measured by the average PI width divided by the marginal coverage rate (the smaller the better). Since TQA-E could create infinite PIs, we replace $\infty$ with 2x the widest finite PI. We also include in the Appendix results on full time series, and with Linear Regression instead of LSTM to show the model-agnostic nature of TQA.

Note that although we use equal-length time series in these experiments, for data with variable lengths (such as CLAIM or MIMIC), one could filter the calibration set before querying the quantile. As long as we assume exchangeability conditioning on length, all theoretical analysis still holds.

---

[5][48] suggests performing Bonferroni correction to jointly cover the entire horizon (all $T$ steps). We explain in the Appendix why this is problematic.

**Results** The results are presented in Tables 2, 3, and 4. In Table 2, we verify that conformal prediction methods - CFRNN (Split), CQRNN, LASplit and TQA- are empirically cross-sectionally valid. The non-conformal methods (QRNN and DPRNN) have unreliable coverage. In Table 3, we show that TQA can greatly improve the (longitudinal) average coverage rate for the worst TS. (TQA-E is consistently better than TQA-B due to the presence of infinitely-wide PIs.) This is also visualized in Figure 3. Note that although CQRNN and LASplit do not perform quantile adjustment, they model uncertainty directly, which also helps improve the longitudinal coverage but is less robust. In Table 4, we verify that TQA did not achieve better coverage simply by using very wide PIs (which is however the case for DPRNN on `GEFCom`). This is somewhat surprising because from Figure 4, the marginal gain in coverage decreases fast as $a_t$ decreases. The PIs for TQA-B should be wider due to the slight over-coverage, and the convexity (with any quantile adjustment). This suggests that TQA-B performs the budgeting very efficiently to cancel out both effects. The efficiency of TQA-E seems low due to the infinitely-wide PIs (replaced by 2x maximum finite width in this computation), but we will see that it generates mostly finite PIs, and the median width is still competitive.

Finally, we would like to also emphasize that any nonconformity scores could theoretically be combined with TQA. In this paper, and in our experiments, we mostly tried to combine the simplest nonconformity scores used in CFRNN (Split) with TQA. The question of how to combine TQA with other nonconformity scores (such as those used in CQRNN or LASplit) is left for future research.

Table 2: Average coverage rate. Empirically valid methods are in **bold** (at p = 0.01). As expected, conformal baselines are valid, while others (QRNN and DPRNN) are not. Note that `GEFCom` does not satisfy the exchangeability assumption, causing invalid coverage for most conformal methods. However, TQA still outperforms all conformal baselines, with TQA-E still valid.

| Coverage | TQA-B | TQA-E | CFRNN (Split) | CQRNN | LASplit | QRNN | DPRNN |
|---|---|---|---|---|---|---|---|
| MIMIC | **91.31±1.32** | **91.19±0.48** | **90.06±1.73** | **90.15±1.24** | **90.33±1.54** | 86.90±1.22 | 46.30±3.84 |
| CLAIM | **91.19±0.49** | **91.56±0.35** | **90.21±0.56** | **90.15±0.68** | **90.20±0.64** | 85.90±0.78 | 24.79±0.85 |
| COVID | **90.79±1.45** | **91.73±0.85** | **90.25±1.69** | **90.08±1.62** | **90.18±1.46** | 89.19±1.54 | 67.51±3.76 |
| EEG | **90.73±1.21** | **90.63±0.75** | **89.92±1.44** | **89.99±1.76** | **89.80±1.15** | 87.96±0.82 | 39.24±1.30 |
| GEFCom | 89.58±0.25 | **90.94±0.14** | 88.61±0.16 | 89.16±0.17 | 88.96±0.18 | 80.40±1.36 | 89.50±0.73 |
| GEFCom-R | **90.56±0.64** | **90.72±0.45** | **89.92±0.78** | **90.07±0.63** | **89.95±0.72** | 85.49±1.08 | **91.03±0.76** |

Table 3: The tail coverage rate (mean longitudinal coverage for the least-covered 10% TS), the higher the better. The best method is in **bold**, and the best method without using any infinitely-wide PI is underscored. Both versions of TQA consistently outperform all baselines.

| Tail Coverage Rate ↑ | TQA-B | TQA-E | CFRNN (Split) | CQRNN | LASplit | QRNN | DPRNN |
|---|---|---|---|---|---|---|---|
| MIMIC | 71.59±4.03 | **80.68±1.74** | 62.22±7.09 | 68.60±3.84 | 65.05±6.12 | 61.80±3.91 | 17.24±5.38 |
| CLAIM | 74.16±1.22 | **81.53±0.77** | 65.95±1.88 | 66.45±3.19 | 68.08±2.44 | 53.89±3.59 | 1.65±0.54 |
| COVID | 70.01±4.45 | **82.39±1.28** | 64.41±6.11 | 66.41±5.99 | 67.38±4.63 | 65.16±6.15 | 36.65±5.63 |
| EEG | 70.99±2.18 | **79.03±1.22** | 64.14±3.42 | 61.95±4.71 | 67.13±2.32 | 57.82±2.78 | 12.99±1.32 |
| GEFCom | 68.96±1.70 | **81.77±0.36** | 58.49±1.38 | 61.63±1.56 | 60.46±1.66 | 47.56±2.27 | 67.45±1.69 |
| GEFCom-R | 75.28±1.28 | **81.80±0.69** | 68.76±2.18 | 71.95±1.66 | 70.79±2.12 | 64.99±1.92 | 71.86±1.75 |

Table 4: Inverse Efficiency, measured by the mean PI width divided by the coverage rate. Since TQA-E can create infinite PI, the width is computed by replacing $\infty$ with 2x the maximum finite PI width. The most efficient (and valid) method is in **bold** (p-value=0.01). As we can see, TQA-B is highly competitive in efficiency.

| Inverse Efficiency ↓ | TQA-B | TQA-E | CFRNN (Split) | CQRNN | LASplit | QRNN | DPRNN |
|---|---|---|---|---|---|---|---|
| MIMIC | 1.990±0.165 | 2.382±0.265 | 1.964±0.170 | **1.738±0.145** | 2.072±0.223 | 1.623±0.146 | 1.258±0.132 |
| CLAIM | 3.020±0.045 | 3.279±0.074 | 3.003±0.052 | **2.902±0.044** | 3.009±0.064 | 2.691±0.035 | 2.401±0.205 |
| COVID | **0.831±0.032** | 1.167±0.337 | **0.826±0.034** | 0.908±0.091 | **0.826±0.037** | 0.888±0.096 | 0.744±0.050 |
| EEG | **1.449±0.025** | 1.749±0.125 | **1.445±0.031** | 1.586±0.052 | **1.448±0.025** | 1.497±0.042 | 1.061±0.027 |
| GEFCom | 0.238±0.005 | 0.280±0.013 | **0.235±0.005** | 0.242±0.005 | 0.238±0.005 | 0.211±0.005 | 0.636±0.009 |
| GEFCom-R | **0.200±0.004** | 0.222±0.010 | **0.198±0.004** | 0.207±0.004 | 0.201±0.004 | 0.193±0.004 | 0.590±0.009 |

## 6 Conclusions

In this paper, we proposed Temporal Quantile Adjustment, or TQA, to quantify uncertainty (create prediction intervals) in time series forecasting with a cross-section. TQA belongs to the framework of conformal prediction, and the main idea is to adjust the quantile to query using temporal information collected so far. This allows TQA to work with *any model and any nonconformity score design*.

TQA theoretically is "no-worse" in cross-sectional coverage than vanilla split conformal as long as the expected value of adjustment is zero, and empirically *improves* the coverage. We also proposed two variants, TQA-B and TQA-E, both of which significantly outperform baselines in improving temporal/longitudinal coverage across many real world datasets. We hope that this work will serve as a foundation for the future design of PIs with both high cross-sectional and temporal coverage.

## 7   Acknowledgment

This work was supported by NSF award SCH-2205289, SCH-2014438, IIS-1838042, NIH award R01 1R01NS107291-01.

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
