# A   Proofs

## A.1   Proof for Theorem 4.1

*Proof.* Denote $\hat{r}_{N+1,t} = 1 - \alpha + \hat{\delta}_{N+1,t}$. To simplify the notation, in the following, all $\mathbb{P}$ means $\mathbb{P}_{\mathbf{s}_{N+1} \sim \mathcal{P}_S}$. We have:

$$\mathbb{P}\{Y_{N+1,t} \in \hat{C}_{a_{N+1,t}, N+1,t}\} \geq \mathbb{P}\{rank_{N+1,t} \leq \hat{r}_{N+1,t}\} \tag{11}$$

$$= \int_0^1 \int_0^q f_{rank_{N+1,t}, \hat{r}_{N+1,t}}(r, q) dr dq \tag{12}$$

$$= \int_0^1 \int_0^q f_{rank_{N+1,t}}(r) f_{\hat{r}_{N+1,t}}(q) dr dq \tag{13}$$

$$= \int_0^1 q f_{\hat{r}_{N+1,t}}(q) dq \tag{14}$$

$$= \mathbb{E}[\hat{r}_{N+1,t}] = 1 - \alpha + \mathbb{E}[\hat{\delta}_{N+1,t}] \tag{15}$$

$\square$

## A.2   Proof for Theorem 4.2

*Proof.* For any $t$, we know that $\hat{r}_{N+1,t}$ follows a uniform distribution among $\{\frac{j}{N}\}_{j=0}^N$ by exchangeability. For simplicity, we drop the subscripts $\cdot_{N+1,t}$. As a result, we have

$$\mathbb{E}[\hat{r}|\hat{r} \geq 1 - \alpha] = \frac{\lceil(1-\alpha)N\rceil + N}{2N} \tag{16}$$

$$\mathbb{E}[\hat{r}|\hat{r} < 1 - \alpha] = \frac{\lceil(1-\alpha)N\rceil - 1}{2N} \tag{17}$$

More over,

$$\mathbb{P}\{\hat{r} \geq 1 - \alpha\} = 1 - \frac{\lceil(1-\alpha)N\rceil}{N+1} \tag{18}$$

$$\mathbb{P}\{\hat{r} < 1 - \alpha\} = \frac{\lceil(1-\alpha)N\rceil}{N+1}. \tag{19}$$

Denoting $\epsilon := \alpha N - \lfloor \alpha N \rfloor = \lceil(1-\alpha)N\rceil - (1-\alpha)N$, this means

$$\mathbb{E}[g^B(\hat{r}; \alpha)] = \left(1 - \frac{\lceil(1-\alpha)N\rceil}{N+1}\right)\left(\frac{\lceil(1-\alpha)N\rceil + N}{2N} - (1-\alpha)\right) \tag{20}$$

$$+ C\frac{\lceil(1-\alpha)N\rceil}{N+1}\left(\frac{\lceil(1-\alpha)N\rceil - 1}{2N} - (1-\alpha)\right) \tag{21}$$

$$= \left(1 - \frac{(1-\alpha)N + \epsilon}{N+1}\right)\left(\frac{(2-\alpha)N + \epsilon}{2N} - (1-\alpha)\right) \tag{22}$$

$$+ C\frac{(1-\alpha)N + \epsilon}{N+1}\left(\frac{(1-\alpha)N - 1 + \epsilon}{2N} - (1-\alpha)\right) \tag{23}$$

$$= \left(\frac{\alpha N + 1 - \epsilon}{N+1}\right)\left(\frac{\alpha N + \epsilon}{2N}\right) \tag{24}$$

$$+ C\frac{(1-\alpha)N + \epsilon}{N+1}\left(\frac{(\alpha-1)N - 1 + \epsilon}{2N}\right) \tag{25}$$

$$= \frac{\alpha^2 N^2 + \alpha N + \epsilon - \epsilon^2}{2(N+1)N} - C\frac{(1-\alpha)^2 N^2 + (1-\alpha)N + \epsilon - \epsilon^2}{2(N+1)N} \tag{26}$$

$$\tag{27}$$

As a result,

$$\mathbb{E}[g^B(\hat{r};\alpha)] = 0 \Leftrightarrow C = \frac{(\alpha N + \epsilon)(\alpha N + 1 - \epsilon)}{((1-\alpha)N + \epsilon)((1-\alpha)N + 1 - \epsilon)} \tag{28}$$

$\square$

## A.3 Proof for Theorem 4.3

*Proof.* The inequality trivially holds for $\alpha \geq 0.5$, so we only focus on the case when $\alpha < 0.5$.

An upper-bound for the additional loss is simply $L := -\min_q g^B(q;\alpha)$. To find an upper bound for $L$, notice that

$$L = -\min_q g^B(q;\alpha) = (1-\alpha)C \tag{29}$$

$$= \frac{(\alpha N + \epsilon)(\alpha N + 1 - \epsilon)}{((1-\alpha)N + \epsilon)((1-\alpha)N + 1 - \epsilon)}(1-\alpha) \tag{30}$$

$$= \frac{\alpha^2 N^2 + \alpha N + \epsilon - \epsilon^2}{(1-\alpha)^2 N^2 + (1-\alpha)N + \epsilon - \epsilon^2}(1-\alpha) \tag{31}$$

$$= \left(1 - \frac{N^2[(1-\alpha)^2 - \alpha^2] + N(1-2\alpha)}{(1-\alpha)^2 N^2 + (1-\alpha)N + \epsilon - \epsilon^2}\right)(1-\alpha) \tag{32}$$

If we consider the last expression as a function of $\epsilon$, it is maximized with $\epsilon = \frac{1}{2}$, which means

$$L \leq \left(\frac{\alpha + \frac{1}{2N}}{1 - \alpha + \frac{1}{2N}}\right)^2 (1-\alpha) \tag{33}$$

Denote $\alpha^+ := \alpha + \left(\frac{\alpha + \frac{1}{2N}}{1 - \alpha + \frac{1}{2N}}\right)^2 (1-\alpha)$. We have $\hat{C}^{split}_{\alpha^+, N+1, t+1} \subseteq \hat{C}^{TQA-B}_{\alpha, N+1, t+1}$. This means

$$\mathbb{P}_{\mathbf{S}_{N+1} \sim \mathcal{P}_S}\{Y_{N+1,t+1} \in \hat{C}^{TQA-B}_{\alpha, N+1, t+1}\} \tag{34}$$

$$\geq \mathbb{P}_{\mathbf{S}_{N+1} \sim \mathcal{P}_S}\{Y_{N+1,t+1} \in \hat{C}^{split}_{\alpha^+, N+1, t+1}\} \tag{35}$$

$$\geq 1 - \alpha - \left(\frac{\alpha + \frac{1}{2N}}{1 - \alpha + \frac{1}{2N}}\right)^2 (1-\alpha) \tag{36}$$

$\square$

## A.4 Proof for Theorem 4.4

*Proof.* First, let us consider the alternative quantile adjustment $b^{TQA-E}_{N+1,t}$ with update rule:

$$b_{t+1} \leftarrow \begin{cases} b_t + \gamma(\alpha - err_t) & (b_t \in [0,1]) \\ b_t + \gamma(\alpha - b_t) & (otherwise) \end{cases}. \tag{37}$$

We will first show $\forall t, \mathbb{E}[b_t] = \alpha$ by induction, as we hinted in the main paper. (For simplicity in notation, we drop the subscript $\cdot_{N+1}$.) To begin with, we have

$$\mathbb{E}_{\mathbf{S}}[b^{TQA-E}_0] = \mathbb{E}_{\mathbf{S}}[\alpha] = \alpha \tag{38}$$

Now,

$$\mathbb{E}_{\mathbf{S}}[b_{t+1}|b_t] = \begin{cases} b_t + \gamma(\alpha - \mathbb{E}_{\mathbf{S}}[err_t]) & (b_t \in [0,1]) \\ b_t + \gamma(\alpha - b_t) & (otherwise) \end{cases} \tag{39}$$

If the nonconformity score's rank has no temporal dependence, we have $err_t \sim Bernoulli(a_t)$ (here again we assume we use random coin-flips to ensure an exact coverage probability of $1 - \alpha$), which means $\mathbb{E}[err_t] = b_t$. This leads to

$$\mathbb{E}_{\mathbf{S}}[b_{t+1}|b_t] = b_t + \gamma(\alpha - b_t) \tag{40}$$

$$\implies \mathbb{E}_{\mathbf{S}}[b_{t+1}] = \mathbb{E}_{\mathbf{S}}[\mathbb{E}_{\mathbf{S}}[b_{t+1}|b_t]] = (1-\gamma)\mathbb{E}_{\mathbf{S}}[b_t] + \gamma\alpha = \alpha \tag{41}$$

By induction, we have

$$\forall t, \mathbb{E}_{\mathbf{S}}[b_t] = \alpha \tag{42}$$

Next, we should compare $\mathbb{E}[a_t]$ and $\mathbb{E}[b_t]$. It should be clear that

$$\forall c, \mathbb{E}_{\mathbf{S}}[a_{t+1}|a_t = c] \leq \mathbb{E}_{\mathbf{S}}[b_{t+1}|b_t = c]. \tag{43}$$

To begin with, we have $\mathbb{E}[a_0] \leq \alpha$. Following a similar logic, we have

$$\mathbb{E}_{\mathbf{S}}[a_{t+1}] = \mathbb{E}_{\mathbf{S}}[\mathbb{E}_{\mathbf{S}}[a_{t+1}|a_t]] \leq (1 - \gamma)\mathbb{E}_{\mathbf{S}}[a_t] + \gamma\alpha \leq \alpha \tag{44}$$

By induction, we get

$$\forall t, \mathbb{E}_{\mathbf{S}_{N+1} \sim \mathcal{P}_S}[a_{N+1,t}^{TQA-E}] \leq \alpha. \tag{45}$$

$\square$

**Remarks** The alternative update rule in Eq. 37 is not just used to prove Theorem 4.4. In fact, one might want to use this rule in practice if the TS is not very long and the asymptotic guarantee (Theorem 4.5) is not relevant. This is because Eq. 37 does not become more conservative on average as $t$ increases.

## A.5    Proof for Theorem 4.5

*Proof.* We will use an similar argument similar to one in [17]. First, we have $a_t \in [-\gamma, 1 + \gamma]$ if $a_0 = \alpha$. To see this, note that $a_{t+1} > a_t \implies a_t \leq 1 \implies a_{t+1} \leq 1 + \gamma$, which means $\forall t, a_t \leq 1 + \gamma$. Similarly, $a_{t+1} < a_t \implies a_t \geq 0 \implies a_{t+1} \geq -\gamma$, which means $\forall t, a_t \geq -\gamma$.

Note that with Eq. 9, we have $a_{t+1} \leq a_t + \gamma(\alpha - a_t)$. Now, if we expand $a_T$, we get

$$-\gamma \leq a_T \leq \alpha + \sum_{t=0}^{T-1} \gamma(\alpha - err_t) \tag{46}$$

$$\implies \frac{\sum_{t=0}^{T-1} err_t}{T} \leq \alpha + \frac{\alpha + \gamma}{\gamma T} \tag{47}$$

By taking the limit of both sides, we are done.

$\square$

# B  Additional Experiment Details

## B.1  Datasets

MIMIC [23, 18, 22]: MIMIC-III is a large public (but requires application of access) database consisting of records for patients admitted to critical care units. The task is to predict white blood cell counts (WBCC) levels, using 25 features (following [1]) including measurements like systolic/diastolic blood pressure or the dosage of antibiotics. To prepare the data, we follow [48] and restrict our study to patients on antibiotics (Levofloxacin). We drop sequences with less than 30 visits, ending up with 392 sequences in total. The exact SQL queries and data processing scripts are provided in the supplemental material and will be published. Note that the EHR has been de-identified - in particular, all visits for a patient are shifted in dates, so we only know the relative dates of visits belonging to the same patient, but do not know the actual dates, making it impossible to perform any temporal splitting.

CLAIM: CLAIM is a proprietary dataset from a large healthcare data provider in North America. It contains longitudinal view of information like inpatient and outpatient services, prescription, costs or enrollment. The task is to predict the (allowed) claim amount for the next visit. The features include diagnoses codes (ICD-10) and procedure codes (CPT/HCPCS), the gender and age of patient, and the previous claim amount.

COVID [10]: COVID is available at https://coronavirus.data.gov.uk/. We follow the setup of [48] and treat different regions within UK as the cross-section, and we used the most recent full-month data available at the time of this work (March 2022). The feature only consists of previous day's case number. We include COVID for completeness as it was used in [48], and as an idealized experiment, despite its unrealistic settings.

EEG [51]: EEG is available at https://archive.ics.uci.edu/ml/datasets/EEG+Database. The "Large" version of the dataset contains the electroencephalography (EEG) signals sampled at 256 Hz for 1 second for 10 alcoholic and 10 control subjects. Like [48], we used only the control group, and downsampled the signals to facilitate training, but to length 64 instead of 50 to avoid unnecessary interpolation. We do not mix data from different EEG channels, and use only channel 0 data because different EEG channels should follow different developments. The feature only consists of the EEG signal at the previous step.

GEFCom [20]: GEFCom (2014) is the Probabilistic Electric Load Forecasting task in Global Energy Forecasting Competition 2014. It has hourly temperature and electricity load data for one utility. The load data covers 9 years. For this task, we split the data into 24-hour-sequences, and consider these sequences as forming a cross-section. The features consist of temperature data from 25 stations, and the previous load.

**Data Licenses and Consent**:

- MIMIC: The PhysioNet Credentialed Health Data License. The data is public, but requires access application.
- COVID: Open Government Licence v3.0. Publicly available on https://coronavirus.data.gov.uk/.
- CLAIM: This is proprietary data.
- EEG: There are no usage restrictions on this data per https://archive.ics.uci.edu/ml/datasets/eeg+database.
- GEFCom: We could not find the license, but it is publicly available (provided by the author) on http://blog.drhongtao.com/2017/03/gefcom2014-load-forecasting-data.html.

## B.2  Implementation and Experiment Details

We use mostly the same architecture and training protocols as [48]. The RNN (mean estimator) is a one-layer LSTM [19], with an embedding size of 32. The optimizer is ADAM [24], and the learning rate is 1e-3. The training epochs for MIMIC/CLAIM/COVID/EEG/GEFCom are 200/500/1000/100/1000. We use the same settings for the residual predictor (for LASplit) and QRNN, except that

- The target for the residual predictor is replaced by $|y - \hat{y}|$.
- The loss for QRNN is replaced with the quantile loss, and the output is two scalars instead of one.

We normalize each feature and $Y$ by the mean and standard deviation in order for the underlying LSTM to train. In each repetition of a experiments, we pool all data together and re-split them into training/calibration/test set using a new seed, and the LSTMs are also trained using different seeds each time. The only exception is `GEFCom` (the non-R version), which has only one data splitting due to the temporal order. The LSTMs for `GEFCom` are still trained using different seeds. All code is provided in the supplemental material and will be published after the review period. Experiments in this paper are fast to run on a personal laptop - the largest single experiment is for `CLAIM` (without repetition), which takes about half an hour, and all experiments finish within 2 hours (for one seed).

## C  Variants of TQA-B

### C.1  Rank-based Quantile Prediction

Instead of the quantile prediction we used in the paper, we also tried the following prediction. $\hat{r}_{i,t+1}^{ewa} = Q^{-1}(\overline{r}_{i,t}; \{\overline{r}_{j,t}\}_{j=1}^{N+1})$ where $\overline{r}_{i,t} := \frac{\sum_{t'=1}^{t} \beta^{t+1-t'} r_{i,t'}}{\sum_{t'=1}^{t} \beta^{t+1-t'}}$. This is an exponentially weighted (decaying) average of the past ranks. We use $\beta = 0.8$, and refer to such quantile prediction as **Rank**-based (as opposed to **Scaled**-based in Section 4.2).

### C.2  The Aggressive Quantile Budgeting

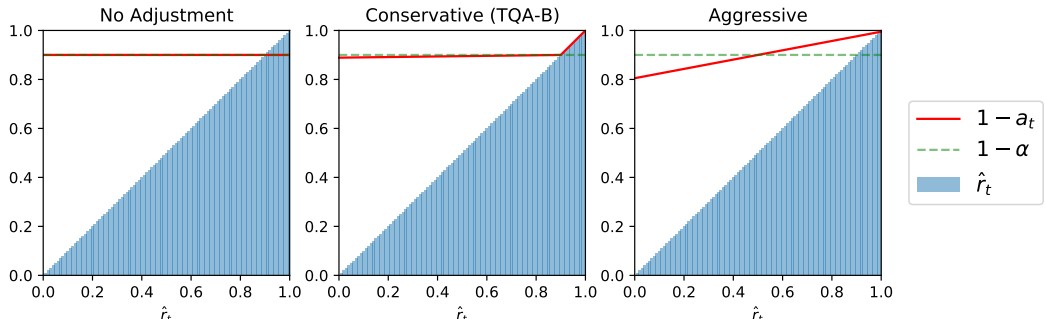

Figure 5: Here we present visualizations of different quantile budgeting methods. "Budgeting" refers to the fact that all red lines are on average $1 - \alpha$. The first picture (No Adjustment) is the same as the basic split conformal method. The second picture refers to the quantile budgeting method presented in Section 4.2. It is conservative because even for a very small $\hat{r}_t$ like 0, the adjusted $a_t$ is still close to $\alpha$. The third picture is the more aggressive budgeting method, and it could set $a_t$ to as big as $2\alpha$. In a sense, it trusts the prediction $\hat{r}_t$ and adapts to it more. As a result, if $\hat{r}_t$ predicts the actual rank of nonconformity scores well, it can achieve a much higher coverage. The extreme case is if $\hat{r}_t$ perfectly predicts the realized ranks: "No Adjustment" will still have $1 - \alpha$ coverage whereas the other two will achieve perfect coverage.

In this section, we present an alternative quantile budgeting function to $g^B$. The specific adjustment is given by

$$g^{BC}(q; \alpha) := \frac{q - 0.5}{2\alpha}. \tag{48}$$

Like in Section 4.2, one could use a $\lambda$ to ensure the final $a_t$ is not too close to zero (to avoid infinitely-wide PIs). We refer to this "aggressive" version as TQA-BC. Figure 5 visualizes the difference between "conservative" (TQA-B) and "aggressive" budgeting. It should be clear that TQA-BC also satisfies Theorem 4.2, Theorem 4.3, however, becomes looser:

**Theorem C.1.**

$$\mathbb{P}_{\mathbf{S}_{N+1} \sim \mathcal{P}_S}\{Y_{N+1,t+1} \in \hat{C}_{\alpha,N+1,t+1}^{TQA-BC}\} \geq 1 - 2\alpha. \tag{49}$$

The proof is essentially the same as that for Theorem 4.3.

## C.3 Comparison

The full results of different variants of TQA-B are in Table 5. All variants are generally valid (except for a tiny drop in coverage for Rank+Conservative on GEFCom-R). In general, among the two quantile prediction methods, Scale tends to produce better tail coverage rate, at the cost of worse efficiency. Among the quantile budgeting methods, the Aggressive version also tends to increase tail coverage rate and decrease efficiency. In practice, one might design the two components depending on the whether efficiency or tail coverage rate is more important.

Table 5: Mean coverage, tail coverage, and inverse efficiency for different variants of TQA-B. Valid mean coverage and the best of tail coverage and inverse efficiency are in **bold**. Note that we re-scaled all methods' mean PI width to the same as CFRNN for the TCR here, for a fair comparison. (We couldn't do this in a meaningful way for TQA-E in the main text.)

| | TQA-B | | | | CFRNN (Split) |
|---|---|---|---|---|---|
| Quantile Prediction Method | Scale | | Rank | | |
| Budgeting Method | Conservative | Aggressive | Conservative | Aggressive | |
| **Coverage** | | | | | |
| MIMIC | **91.31±1.32** | **93.62±1.04** | **89.79±1.68** | **91.79±1.41** | **90.06±1.73** |
| CLAIM | **91.19±0.48** | **92.87±0.37** | **89.88±0.55** | **91.54±0.46** | **90.21±0.56** |
| COVID | **90.79±1.45** | **92.46±1.15** | **89.94±1.71** | **91.16±1.44** | **90.25±1.69** |
| EEG | **90.73±1.21** | **92.53±1.00** | **89.57±1.46** | **90.95±1.28** | **89.92±1.44** |
| GEFCom | 89.58±0.25 | **91.19±0.39** | 87.96±0.20 | 89.62±0.22 | 88.61±0.16 |
| GEFCom-R | **90.56±0.64** | **91.52±0.57** | 89.55±0.80 | **90.57±0.67** | **89.92±0.78** |
| **Tail Coverage Rate (rescaled) ↑** | | | | | |
| MIMIC | 70.18±4.17 | **76.84±4.20** | 63.06±6.46 | 70.98±4.69 | 62.22±7.09 |
| CLAIM | 73.36±1.27 | **78.15±1.20** | 66.73±1.76 | 73.08±1.29 | 65.95±1.88 |
| COVID | 69.48±4.48 | **75.49±4.89** | 64.85±5.80 | 69.44±5.51 | 64.41±6.11 |
| EEG | 70.29±2.15 | **75.81±2.69** | 64.50±3.32 | 69.51±3.01 | 64.14±3.42 |
| GEFCom | 67.71±1.68 | **74.19±2.55** | 58.51±1.27 | 67.52±1.42 | 58.49±1.38 |
| GEFCom-R | 74.48±1.36 | **77.47±1.36** | 69.18±2.14 | 73.59±1.58 | 68.76±2.18 |
| **Inverse Efficiency ↓** | | | | | |
| MIMIC | **1.990±0.165** | 2.128±0.182 | **1.936±0.166** | **1.979±0.166** | **1.964±0.170** |
| CLAIM | 3.020±0.045 | 3.224±0.072 | **2.956±0.051** | 3.025±0.046 | 3.003±0.052 |
| COVID | **0.831±0.032** | 0.858±0.036 | **0.819±0.033** | **0.828±0.033** | **0.826±0.034** |
| EEG | 1.449±0.025 | 1.498±0.023 | **1.429±0.029** | 1.444±0.025 | 1.445±0.031 |
| GEFCom | 0.238±0.005 | 0.247±0.007 | **0.233±0.005** | **0.236±0.005** | **0.235±0.005** |
| GEFCom-R | 0.200±0.004 | 0.210±0.006 | **0.196±0.004** | 0.199±0.004 | 0.198±0.004 |

# D   Additional Results

**Full Time Series**: We repeat the experiments in Section 5 but measure the evaluation metrics on the entire time series as opposed to the last 20 steps. The results are in Table 6 and the conclusion stays the same.

**Percentage of Infinitely-wide PIs**: Since TQA-E could generate infinitely-wide PIs, we check how many such intervals are actually created. The results are in Table 7, and it shows that most $\hat{C}^{TQA-E}$ are finite.

**Median and Mean Width of the PIs**: We examine the raw width of the PIs. Note that because TQA-B tends to improve the coverage slightly, the width is slightly higher. In addition, the mean width should be even higher due to the attempt to cover extreme residuals (as discussed/showed in Figure 4). However, from Table 8, we note that the difference in the raw PI widths are also small.

**CFRNN with Bonferroni Correction** (CFRNN-Bon): As explained in Section 3.2, to perform the correct split conformal prediction, we need to use $\infty$ in place of $v_{N+1}$ (in order to avoid plugging in different $y$ values, usually defined on a fine-granular grid, which is very expensive). This means that, if we perform Bonferroni Correction as proposed in [48] by querying the $1 - \frac{\alpha}{T}$ instead of $1 - \alpha$, the PI will *always* by infinitely wide if $\frac{\alpha}{T} <= \frac{1}{N+1}$. In our experiments, this is the case for MIMIC, COVID, EEG and GEFCom/GEFCom-R. The original paper [48] implemented the split-conformal incorrectly by ignoring $v_{N+1}$, which is why this issue did not appear. In addition, even if we ignore the issue of (constant) infinitely-wide PIs, in our settings the length of the time-series (e.g. number of

Table 6: Mean coverage, tail coverage, and inverse efficiency using the entire time series (as opposed to the last 20 steps). Valid mean coverage and the best of tail coverage and inverse efficiency are in **bold**. The conclusion is the same as in the main text - TQA greatly improves the longitudinal coverage for the least-covered TSs, and in particular TQA-B maintains very competitive efficiency.

| Coverage | TQA-B | TQA-E | CFRNN (Split) | CQRNN | LASplit | QRNN | DPRNN |
|---|---|---|---|---|---|---|---|
| MIMIC | **91.16±1.15** | **91.76±0.63** | **90.03±1.49** | **90.06±1.17** | **90.19±1.32** | 84.81±1.13 | 44.78±3.61 |
| CLAIM | **91.00±0.37** | **91.38±0.23** | **90.13±0.48** | **90.07±0.54** | **90.11±0.44** | 86.38±0.62 | 25.31±0.72 |
| COVID | **90.70±1.39** | **91.73±0.74** | **90.18±1.61** | **90.08±1.44** | **90.15±1.39** | 89.11±1.45 | 65.40±2.74 |
| EEG | **90.56±0.85** | **91.04±0.30** | **89.82±1.07** | **89.99±1.14** | **89.82±0.96** | 86.82±0.62 | 35.35±1.18 |
| GEFCom | 89.34±0.23 | **90.62±0.12** | 88.50±0.15 | 89.05±0.16 | 88.83±0.16 | 80.58±1.29 | **90.46±0.73** |
| GEFCom-R | **90.56±0.60** | **90.69±0.39** | **89.95±0.75** | **90.13±0.61** | **89.97±0.66** | 85.86±1.08 | **91.97±0.71** |

| Tail Coverage Rate ↑ | | | | | | | |
|---|---|---|---|---|---|---|---|
| MIMIC | 75.22±3.08 | **84.64±0.95** | 67.54±4.99 | 71.42±3.58 | 69.76±4.47 | 62.53±3.73 | 21.16±4.37 |
| CLAIM | 76.26±0.91 | **84.60±0.39** | 68.76±1.71 | 69.75±2.50 | 71.21±1.87 | 59.78±2.85 | 4.55±0.56 |
| COVID | 70.47±4.65 | **84.21±0.89** | 65.22±5.83 | 68.70±5.71 | 69.07±4.59 | 67.19±5.72 | 40.74±4.24 |
| EEG | 75.80±1.25 | **87.22±0.13** | 70.43±2.17 | 69.94±2.80 | 71.62±1.77 | 65.22±1.93 | 18.86±0.94 |
| GEFCom | 67.84±1.66 | **82.71±0.15** | 58.26±1.20 | 61.72±1.48 | 60.36±1.63 | 48.51±2.06 | 71.45±1.54 |
| GEFCom-R | 75.41±1.26 | **83.00±0.31** | 69.14±2.10 | 72.96±1.57 | 71.51±1.88 | 66.64±1.98 | 75.65±1.71 |

| Inverse Efficiency ↓ | | | | | | | |
|---|---|---|---|---|---|---|---|
| MIMIC | 2.053±0.157 | 2.478±0.333 | 2.018±0.152 | **1.831±0.153** | 2.115±0.185 | 1.676±0.148 | 1.277±0.120 |
| CLAIM | 3.039±0.038 | 3.245±0.061 | 3.019±0.043 | **2.934±0.039** | 3.023±0.051 | 2.740±0.031 | 2.242±0.146 |
| COVID | **0.823±0.031** | 1.112±0.246 | **0.819±0.033** | 0.876±0.073 | **0.817±0.038** | 0.857±0.077 | 0.728±0.047 |
| EEG | **1.374±0.016** | 1.662±0.093 | **1.368±0.022** | 1.494±0.039 | **1.368±0.018** | 1.414±0.033 | 0.985±0.017 |
| GEFCom | 0.222±0.005 | 0.257±0.011 | **0.219±0.004** | 0.225±0.004 | 0.221±0.004 | 0.195±0.004 | 0.618±0.009 |
| GEFCom-R | **0.185±0.003** | 0.203±0.009 | **0.183±0.003** | 0.191±0.004 | 0.186±0.004 | 0.178±0.004 | 0.589±0.009 |

Table 7: The percentage of infinitely-wide PIs created by TQA-E for different datasets. Only a very small percentage of the PIs are infinitely-wide.

| % Infinitely-wide PI | MIMIC | CLAIM | COVID | EEG | GEFCom | GEFCom-R |
|---|---|---|---|---|---|---|
| Last 20 steps | 3.10±1.23 | 2.18±0.36 | 3.40±1.47 | 3.78±0.97 | 3.84±0.17 | 2.31±0.52 |
| Full TS | 2.51±0.99 | 1.72±0.28 | 2.74±1.24 | 3.46±0.78 | 3.22±0.15 | 1.94±0.44 |

visits by a patient) might not be known in advance. As a result, it is not clear how to perform the Bonferroni Correction.

**Tail coverage rate over time** Figure 6 shows the tail coverage rate at different values of $t$. Ideal means the coverage is independent for different $t$ within each time series. In general TQA improves the tail coverage noticeably, but there is still gap between Ideal and TQA. TQA-E sometimes approaches this optimal case (at the expense of creating infinitely-wide PIs).

**Mean (cross-sectional) coverage over time** Figure 7 shows the mean coverage rate for different $t$. As expected, conformal methods show cross-sectional validity, whereas the coverage rate for non-conformal methods vary.

**Linear Regression**: We repeat the experiments in Section 5 but replace the underlying LSTM with Linear Regression (by training one model for each $t$, taking input from up to $t-1$). The CLAIM data is skipped in this case due to difficulty in training. The results are in Table 9 and the conclusion stays the same.

**Examples of Actual PIs** Figure 8 how TQA-B and TQA-E improve the coverage for the least-covered TSs over CFRNN (Split) (as they use the same nonconformity score). The rest of the baselines are in Figure 9. We use random seed 1 and pick the 1-th-least-covered TS (according to the mean coverage across all baselines) from each dataset. In general, TQA would increase the width (by lowering $a_t$) if the first few observations of a TS shows some extremity, which is probably most noticeable in CLAIM. CQRNN and LASplit also adapt to the widths by using different nonconformity scores. Again, we note that the quality of TQA-B might be further improved with better prediction $\hat{r}$.

Table 8: Median and mean width of the PIs. The widths of infinitely-wide $\hat{C}^{TQA-E}$ are replaced with 2x the width of the widest finite PI.

| Median Width ↓ | TQA-B | TQA-E | CFRNN (Split) | CQRNN | LASplit | QRNN | DPRNN |
|---|---|---|---|---|---|---|---|
| MIMIC | 1.741±0.165 | 1.639±0.146 | 1.753±0.169 | 1.443±0.120 | 1.640±0.172 | 1.279±0.116 | 0.493±0.023 |
| CLAIM | 2.667±0.053 | 2.561±0.051 | 2.710±0.060 | 2.628±0.048 | 2.456±0.057 | 2.325±0.032 | 0.434±0.023 |
| COVID | 0.735±0.036 | 0.705±0.035 | 0.739±0.038 | 0.776±0.072 | 0.706±0.034 | 0.757±0.084 | 0.390±0.052 |
| EEG | 1.285±0.044 | 1.207±0.035 | 1.291±0.052 | 1.343±0.069 | 1.228±0.039 | 1.218±0.040 | 0.312±0.013 |
| GEFCom | 0.201±0.004 | 0.196±0.004 | 0.198±0.004 | 0.204±0.004 | 0.191±0.004 | 0.155±0.005 | 0.428±0.006 |
| GEFCom-R | 0.171±0.008 | 0.174±0.005 | 0.170±0.008 | 0.171±0.004 | 0.164±0.005 | 0.149±0.005 | 0.458±0.009 |
| Mean Width ↓ | | | | | | | |
| MIMIC | 1.818±0.157 | 2.189±0.245 | 1.769±0.164 | 1.566±0.132 | 1.872±0.212 | 1.411±0.130 | 0.578±0.031 |
| CLAIM | 2.754±0.048 | 3.002±0.070 | 2.709±0.056 | 2.616±0.049 | 2.714±0.067 | 2.312±0.033 | 0.594±0.042 |
| COVID | 0.755±0.033 | 1.070±0.308 | 0.746±0.038 | 0.818±0.084 | 0.745±0.037 | 0.792±0.088 | 0.503±0.050 |
| EEG | 1.315±0.039 | 1.585±0.111 | 1.299±0.048 | 1.428±0.069 | 1.300±0.037 | 1.317±0.043 | 0.416±0.017 |
| GEFCom | 0.213±0.005 | 0.255±0.012 | 0.208±0.004 | 0.216±0.004 | 0.212±0.005 | 0.170±0.005 | 0.569±0.007 |
| GEFCom-R | 0.181±0.004 | 0.201±0.010 | 0.178±0.005 | 0.187±0.004 | 0.181±0.005 | 0.165±0.005 | 0.537±0.011 |

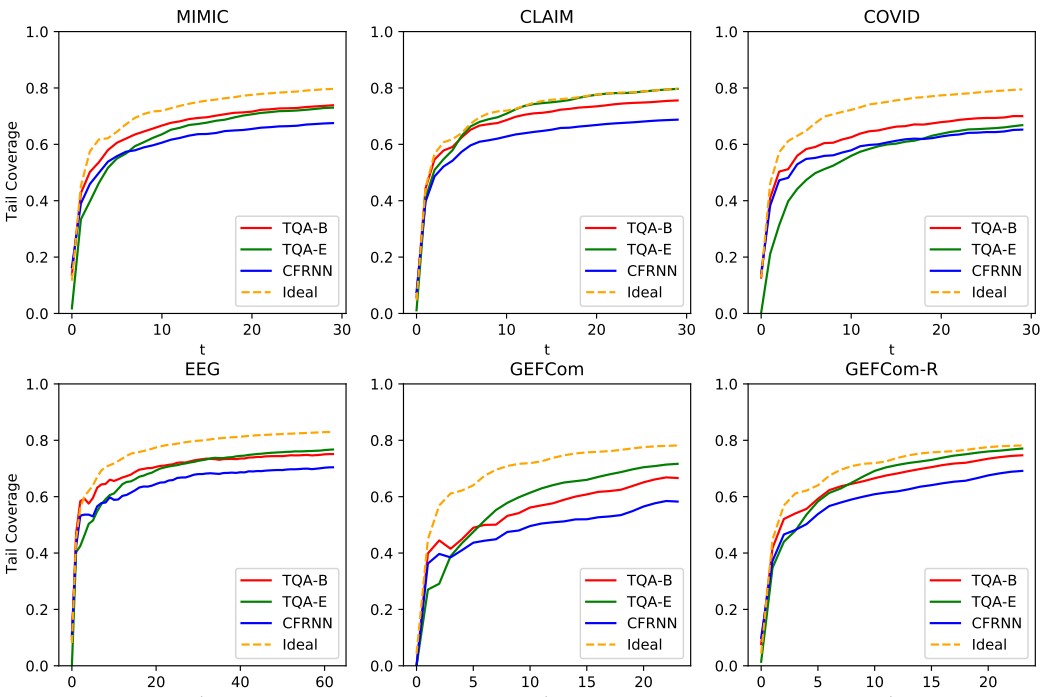

Figure 6: Tail Coverage Rate as a function of time. We plot the mean of 50 experiments. There is, however, still gap between TQA and `Ideal`.

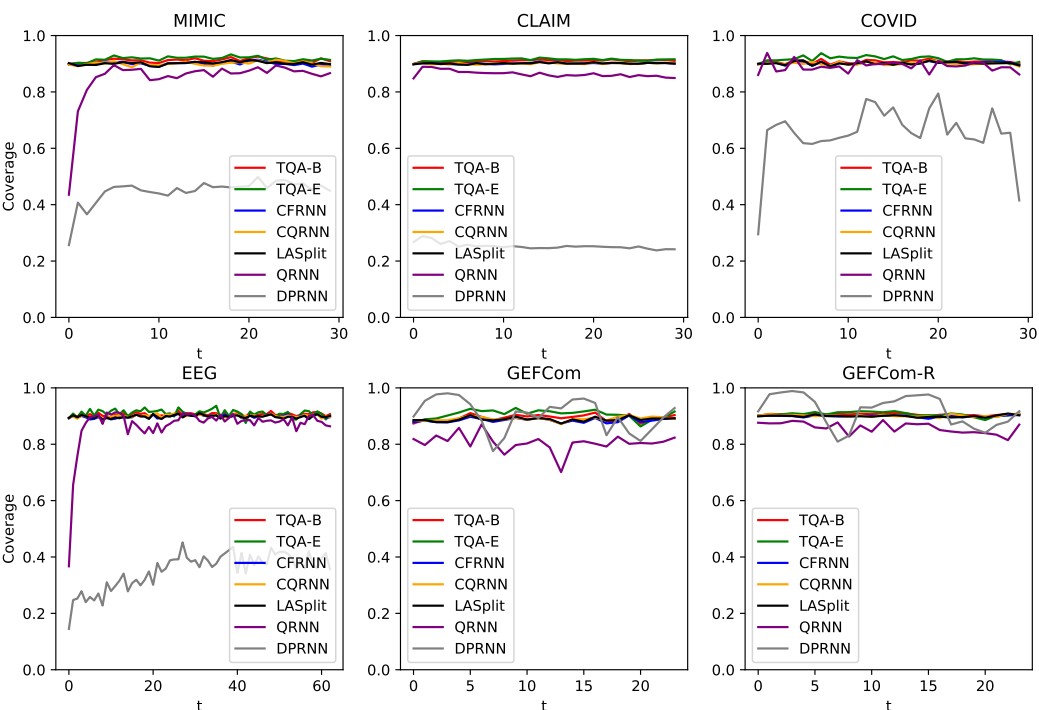

Figure 7: Mean coverage rate at different $t$. Conformal methods are valid at all $t$. Coverage rate for non-conformal methods vary greatly through time.

Table 9: Mean coverage, tail coverage, and inverse efficiency with linear regression being the base point estimator. Valid mean coverage and the best of tail coverage and inverse efficiency are in **bold**. The conclusion is the same as the case of LSTM.

| Coverage | TQA-B | TQA-E | CFRNN (Split) | CQR | LASplit |
|---|---|---|---|---|---|
| MIMIC | **90.65±1.36** | **91.63±0.79** | **89.75±1.61** | **89.96±1.62** | **89.81±1.48** |
| COVID | **90.93±1.45** | **91.95±0.86** | **90.28±1.71** | **90.17±1.57** | **90.25±1.55** |
| EEG | **90.76±1.34** | **90.78±0.84** | **89.92±1.56** | **90.26±1.93** | **90.14±1.32** |
| GEFCom | 89.66±0.17 | **90.46±0.12** | 89.32±0.14 | 88.98±0.28 | 89.21±0.16 |
| GEFCom-R | **90.57±0.63** | **90.78±0.42** | **90.08±0.72** | **90.13±0.66** | **90.12±0.59** |

| Tail Coverage Rate ↑ | | | | | |
|---|---|---|---|---|---|
| MIMIC | 68.80±3.88 | **80.73±1.52** | 60.59±5.64 | 59.40±5.23 | 64.36±4.88 |
| COVID | 70.38±4.04 | **82.20±1.51** | 64.35±6.00 | 68.30±4.75 | 69.41±4.94 |
| EEG | 68.11±2.73 | **77.66±1.27** | 61.09±3.96 | 54.45±6.58 | 66.81±2.70 |
| GEFCom | 73.72±0.92 | **81.58±0.33** | 69.31±0.95 | 70.01±1.12 | 71.40±0.92 |
| GEFCom-R | 75.34±0.88 | **81.61±0.81** | 70.69±1.42 | 72.69±1.39 | 73.01±1.48 |

| Inverse Efficiency ↓ | | | | | |
|---|---|---|---|---|---|
| MIMIC | 3.235±0.263 | 4.092±0.768 | 3.205±0.266 | **2.899±0.252** | 3.326±0.286 |
| COVID | **0.898±0.028** | 1.129±0.171 | **0.893±0.030** | 0.921±0.042 | **0.894±0.028** |
| EEG | **1.840±0.048** | 2.232±0.152 | **1.832±0.057** | 1.976±0.080 | 1.995±0.059 |
| GEFCom | 0.309±0.005 | 0.350±0.007 | **0.307±0.005** | 0.373±0.011 | 0.334±0.006 |
| GEFCom-R | **0.297±0.006** | 0.318±0.009 | **0.296±0.007** | 0.355±0.008 | 0.315±0.006 |

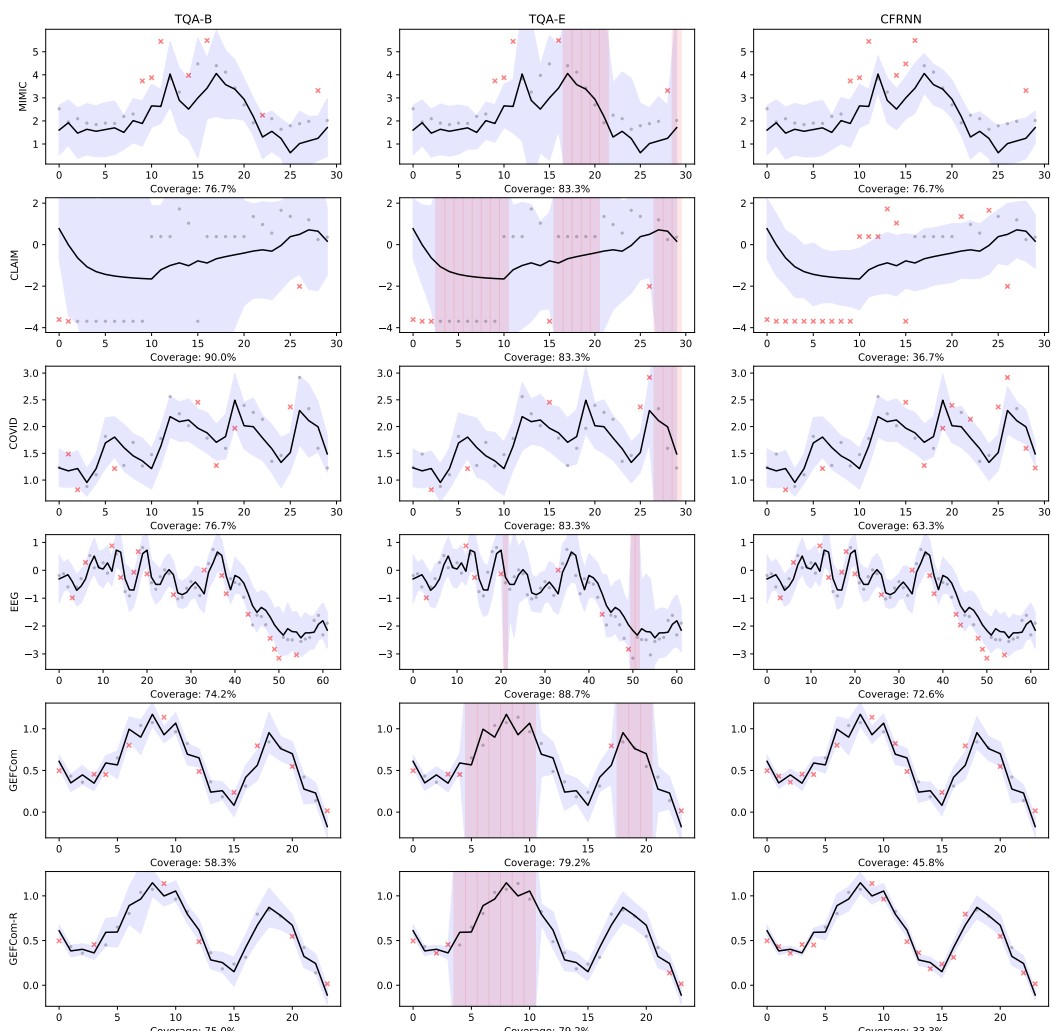

Figure 8: Dark lines are $\hat{y}$, blue bands are the PIs, and × denotes $y$ outside the PI. red bands means infinitely-wide PIs for TQA-E. The adaptive-ness of TQA-B and TQA-E may be most obvious in CLAIM and GEFCom-R. We also note that TQA-E seems to produce quite a few infinitely-wide PIs on these least-covered TSs (somewhat as expected).

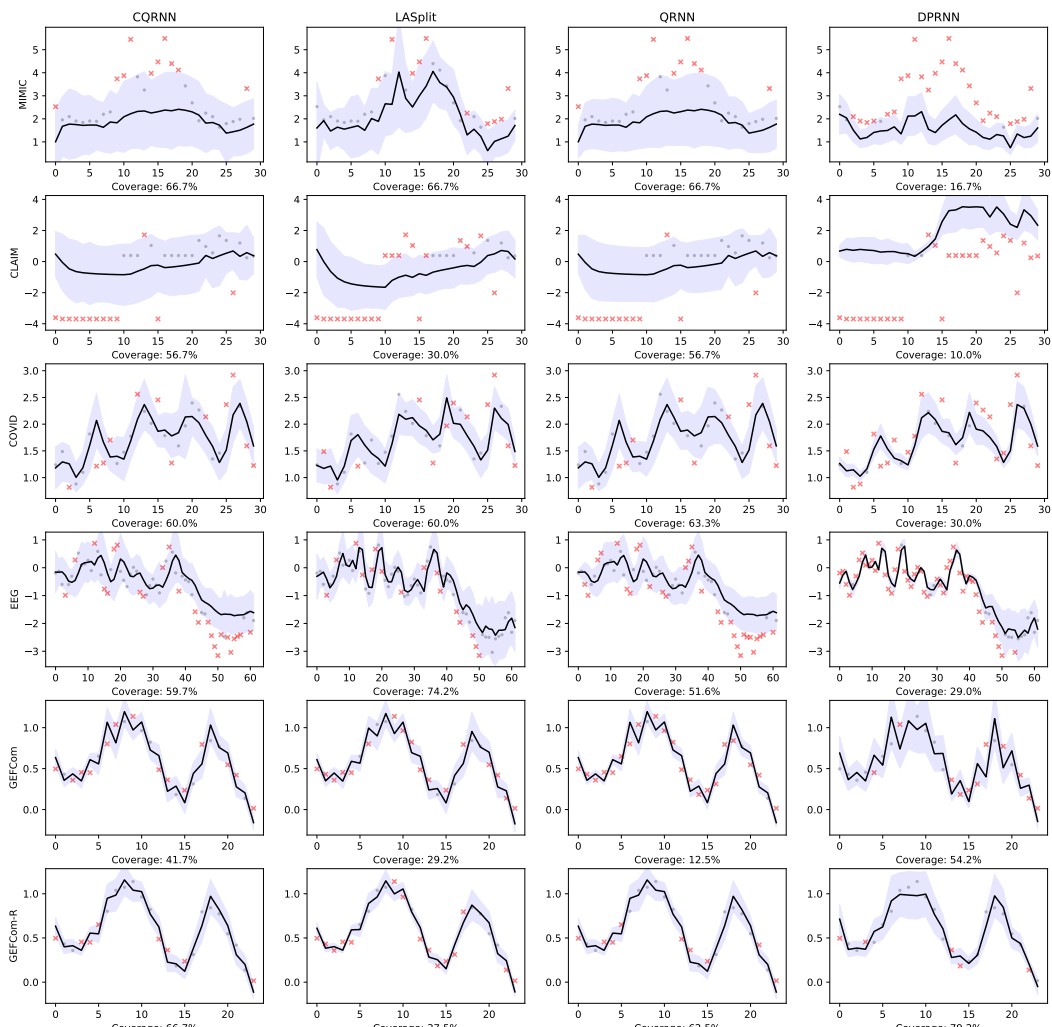

Figure 9: Like Figure 8, but for different baselines.