# OpenReview forum: "Conformal Prediction with Temporal Quantile Adjustments"
_NeurIPS.cc/2022/Conference — NeurIPS 2022 Accept_

### Official Review · Reviewer_HtoR · 2022-07-10

**Rating:** 7
**Confidence:** 1
**Soundness:** 3 good
**Presentation:** 2 fair
**Contribution:** 3 good

**Summary:**

This paper proposes a new approach called temporal quantile adjustments (TQA) that can improve longitudinal coverage and preserve cross-sectional coverage for the prediction interval built for regression on cross-sectional time series data.

The definitions of cross-sectional validity and longitudinal validity of a prediction interval are first presented in Eq (1) and (2). The existing split conformal method is reviewed and it can generate a cross-sectionally valid prediction interval. However, it cannot be directly applied to achieve  longitudinal validity as the exchangeability assumption is not satisfied.

The proposed TQA technique is introduced in section 4. Theorem 4.1 proved the coverage probability is lowered bounded by 1 - \alpha + E[\hat{\delta}_{N+1,t}]. Motivated by Theorem 4.1, TQA-B (quantile budgeting) is introduced in section 4.2 to make the E[\hat{\delta}_{N+1,t}] close to zero. TQA-B consists of 1) quantile prediction, and 2) budgeting. The error-based adjustment (TQA-E) is introduced to increase \delta_{t+1} if there are many errors in the past.

Experimental results show that while existing conformal baseline approaches can achieve cross-sectional validity. TQA-E can achieve longitudinal validity besides cross-sectional validity.

**Questions:**

1) In the definition of longitudinal validity (Eq. 2), the distribution on the RHS is Y_{N+1, t} | S_{N+1,:t-1}, shouldn't it be S_{N+1, t} | S_{N+1,:t-1} instead? Is S_{N+1, t} a random variable in this case?

2) can the author explain why the two-step approach in TQA-B can make E[\hat{\delta}] close to 0?


**Limitations:**

I didn't see major potential negative societal impact of this work.

**Strengths And Weaknesses:**

Strengths
-----------
1) the major contribution of this work, i.e., improving longitudinal coverage and preserve cross-sectional coverage for the prediction interval built for regression on cross-sectional time series data, seems novel.
2) the proposed approach has solid theoretical guarantees.
3) experiment results support the major claim in the paper.

Weaknesses
---------------
I cannot find major weakness

---

> ### Author Response · Authors · 2022-08-02
> **Thank you for the positive feedback**
>
> Thank you for the positive and thorough review.
> We would like to clarify the questions raised below.
>
> > (Eq.2): ... the distribution on the RHS is $Y\_{N+1, t} | S\_{N+1,:t-1}$, shouldn't it be $S\_{N+1, t} | S\_{N+1,:t-1}$ instead?
>
> Here, we assumed that $\hat{C}\_{N+1,t}$ typically takes $X\_{N+1,t}$ as an input as well, so we took it out of the probability subscript.
>
>
> > can the author explain why the two-step approach in TQA-B can make $E[\hat{\delta}]$ close to 0?
>
> $E[\hat{\delta}]$ is set to $0$ by the choice of $C$.
> $E[\hat{\delta}]=0$ in this case (with such $C$) because of assumed data exchangeability. This is proved in details in Appendix A.2. We separate this process into two steps in order to conceptually distinguish them, which is helpful for the theoretical discussion.
> However, jointly learning a budgeting method could have good performance as well, and would be an interesting topic for future research in our opinion.

---

### Official Review · Reviewer_2Rxg · 2022-07-11

**Rating:** 6
**Confidence:** 3
**Soundness:** 3 good
**Presentation:** 2 fair
**Contribution:** 3 good

**Summary:**

Temporal Quantile Adjustment (TQA) is proposed for time-series data with a focus on constructing efficient and valid prediction intervals (PIs) for regressing the models with two distinct notions of coverage: cross-sectional coverage and longitudinal/temporal coverage. TQA adjusts the quantile to query in Conformal Prediction at each time t, accounting for both cross-sectional and longitudinal coverage in a theoretically-grounded manner. The theory part supports the validity of coverages. Two variants are also provided for generality of the TQA, following the nature of quantile adjustment.

**Questions:**

It is probably out of the scope of this paper, but would there be a method to deliver a target coverage when possible?

**Limitations:**



**Strengths And Weaknesses:**

Strength:
- Intends to resolve a critical challenge in conformal prediction, that is to construct efficient valid predictions from both cross-sectional and longitudinal perspectives.
- Clear writing and contribution to the Conformal Prediction problem
- Solid experiment results supporting the methodology in Appendix, Fig 8,9

Weakness:
- lack of guarantee or operational guidance for improving the coverages

---

> ### Author Response · Authors · 2022-08-02
> **Thank you**
>
> Thank you very much for the positive feedback and questions.
>
> ### Strict Longitudinal coverage
>
> Such coverage will require additional assumptions (and they are typically not mild).
> We think (and as the reviewer points out) this is beyond the scope of this paper.
> We do present an asymptotic guarantee in Theorem 4.5 with the help of infinitely-wide intervals.
> We also discuss this in our response to reviewer me7X above.

---

### Official Review · Reviewer_me7X · 2022-07-11

**Rating:** 5
**Confidence:** 4
**Soundness:** 4 excellent
**Presentation:** 4 excellent
**Contribution:** 2 fair

**Summary:**

The paper considers the task of predictive inference with panel data consisting of a time dimension and sample dimension. Two types of coverage guarantees are considered here--- longitudinal coverage and cross-section coverage. In contrast to previous works that focus on either of the coverage guarantees, this paper aims at achieving both (at least empirically). The authors leverage the conformal inference technique, adapting it to the temporal structure. Under certain conditions, the proposed method reserves the cross-section coverage ensured by the conformal inference method; by incorporating the temporal structure, the method(s) achieves better longitudinal coverage empirically.


**Questions:**

1. Line 35: is there a comma missing between $X_{i,t}$ and $Y_{i,t}$

2. Line 36: I assume that the prediction interval also depends on
$X_{N+1, t+1}$ right? Should it be "Given data $[Z_{N+1,j}]^t_{j=1}$ and $X_{N+1, t+1}$"?

3. Line 124:  should the "..." after $Z_{i,T}$ be removed?

4. Line 132: using both the terms "training data" and "training set"
could be confusing.

5. Line 208: is there any reason why $\beta$ is taken to be $0.8$?

6. Line 214: the notation $f(S_{i,:t};[S_{j,:t}]^{N+1}_{j=1})$
is confusing.

7. Line 222: is the quantity $\alpha^{\rm TQA-B}_{N+1,t}$ formally defined?

8. line 293: what does it mean to repeat the prediction 50 times? Is the randomness from sample splitting? If so, how are the time series selected?

9. Fig3: what is the coverage in the y-axis?

10. Appendix A1: $\hat{r}_{N+1,t} = \lceil (1- \alpha + \hat{\delta}_{N+1,t})(N+1) \rceil$?






**Limitations:**

The authors have thoroughly addressed the limitations of their work and potential negative societal impact.

**Strengths And Weaknesses:**

Strength:
1. The paper is very well-written: there is a clear motivation, and a thorough introduction to the related works and their limitations; the proposed method is presented with explanations and intuitions. I enjoyed reading this paper.

2. The research question raised is interesting and can be relevant to practitioners.

3. The proposed method(s) achieves satisfactory empirical performance compared with competitors.

Weaknesses/room for improvement

1. The theoretical guarantee of the proposed method is a bit weak. Even for cross-section coverage, the guarantee (Theorem 4.1) holds under a fairly strong condition (as in the examples given in the remark after Theorem 4.1), although this is mitigated by the lower bound given in Theorem 4.3. There is no guarantee for longitudinal coverage (admittedly, this is impossible w/o further assumptions, but it might be worth understanding it under mild conditions).

2. It might be helpful to present simulation results. With synthetic data, it is possible to directly evaluate the validity of the proposed method and compare it with other competitors.

---

> ### Author Response · Authors · 2022-08-02
> **Thank you for the detailed review**
>
> Thank you for the feedback and for appreciating the soundness and presentation of our paper.
> We would like to clarify on the concerns below.
>
> ### Theoretical guarantee is not strong
>
> For **cross-sectional coverage**, Theorem 4.3 provides a worst case strict coverage guarantee.
> We agree this is less ideal than strict $\geq 1-\alpha$ coverage guarantee, but would like to note that having a small gap is also not unusual (e.g. the guarantee in [1'] is $1-2\alpha$, which is looser than Theorem 4.3).
> This gap technically cannot be improved, as one could carefully design an adversarial relation between $\hat{r}$ and $r$.
> In reality, just like [1'], we do not see such a drop in coverage, but generally see improvement as it is generally unlikely to predict $r$ worse than random guess (which is the assumption of Theorem 4.2).
> We think our theoretical contribution is still significant, as Theorem 4.2 and 4.4 are nontrivial new results despite being intuitive on the surface, and the assumption on the error is not uncommon as well (e.g. [55]).
>
>
> As the reviewer correctly points out,  **longitudinal coverage** is impossible without further assumptions.
> It is not immediately clear to us what constitutes a suitable assumption. Besides, we are focused on proposing a framework to adjust the quantile being queried. Different distributional assumptions could lead to a different ``Quantile Prediction'' step in 4.2.i. As a result, we mostly use experiments to show improvement on longitudinal coverage. It is surely an interesting direction to investigate what is the most practical quantile regression and adjustment method, and what are the corresponding general/mild assumptions.
>
>
> [1'] Barber, Rina Foygel, et al. "Predictive inference with the jackknife+." The Annals of Statistics 49.1 (2021): 486-507.
>
> ### Simulation Results
>
> While it is possible to compute the exact coverage probability of different methods in a simulation, designing such synthetic data unavoidably introduces much bias. We opted for various real-world datasets so as to avoid such bias. We repeated the experiments with 50 random seeds in order to make better comparisons across different methods as well.
>
> ### Questions
>
> Thank you for pointing out typos, which we have of course corrected, and updated the wordings in L35,L36, and L132. We would like to clarify a few items below.
>
> - L124: We allowed the time series to have variable lengths here, as L125 handles the case of up to any finite time $t$.
>
> - L214: This means $f$ can depend on the **set** of all (calibration) time-series as well.
> This includes operation like "taking the rank" of a residual $i$ among the entire calibration set at the same $t$.
>
> - L218: We only tried $\beta=0.8$ and did not tune it.
> The conclusion is similar if we use $\beta=1$ (i.e. no decay):
> |    TQA-B($\beta$=1)    | Coverage      | Tail Coverage | Inverse Efficiency |
> | ------ | ------------- | ------------- | ------------------ |
> | MIMIC  | 90.81$\pm$1.51| 67.59$\pm$5.02| 1.993$\pm$0.171    |
> | CLAIM  | 91.00$\pm$0.56| 71.96$\pm$1.51| 3.014$\pm$0.050    |
> | COVID  | 90.78$\pm$1.57| 69.01$\pm$4.88| 0.831$\pm$0.032    |
> |EEG     | 90.66$\pm$1.38| 67.97$\pm$3.46| 1.456$\pm$0.031    |
> |GEFCom  | 89.51$\pm$0.20| 68.30$\pm$1.03| 0.238$\pm$0.005    |
> |GEFCom-R| 90.45$\pm$0.64| 74.73$\pm$1.17| 0.199$\pm$0.004    |
>
> - L222: $a_{N+1,t}^{TQA-B}$ was implicitly defined with L181, and we **edited the sentence before Theorem 4.2** to explain this.
>
> - L293: The RNN is trained with different seeds, and except for GEFCom (see L280), the time series are first permuted randomly and then split.
>
> - Fig 3: Each point's vertical value means this time series' average coverage rate over time.
> Fig 3 is essentially un-aggregating the tail coverage rate (meaning that the average y-value of the curve corresponds to values in Table 3).
>
> - Appendix A1: No, as $r$, $\alpha$ and $\delta$ are all between 0 and 1 (see Eq. (6)).
> $r$ represents a rank percentile for convenience of discussion.

---

### Official Review · Reviewer_QQnG · 2022-07-11

**Rating:** 8
**Confidence:** 5
**Soundness:** 3 good
**Presentation:** 3 good
**Contribution:** 4 excellent

**Summary:**

The paper develops methods to construct predictive intervals on regression for cross-sectional time series data that come with valid coverage guarantees. The method developed in the paper focus on two types of validity: cross-sectional validity  where the randomness Is the randomness over time series, and longitudinal validity where the randomness is time series specific and is typically temporal dependent. While cross-sectional validity can be achieved following the standard conformal inference approach, achieving the longitudinal validity is the harder one. The paper suggests temporal quantile adjustment to achieve longitudinal validity via quantile budgeting (TQA-B) and error based adjustment (TQA-E). The authors give error bounds for coverage and also support their claim empirically.

**Questions:**

I do not have additional questions.

**Limitations:**

Yes.

**Strengths And Weaknesses:**

Strengths:

1) The underlying problem studied in the paper is extremely relevant to the machine learning community.
2) The paper is supported both theoretically and experimentally showing efficacy of the methods developed.
3) Overall the paper is well written.
4) The ideas developed in the paper are novel.

---

> ### Author Response · Authors · 2022-08-02
> **Thank you very much for the positive feedback**
>
> Thank you very much for the positive feedback on our work. Please do let us know if any questions arise.

---

### Meta-Review · Area_Chair_USea · 2022-08-28

**Recommendation:** Accept
**Confidence:** Certain

**Metareview:**

In this paper, the authors propose the temporal quantile adjustments (TQA) that can improve longitudinal coverage and preserve cross-sectional coverage for the prediction interval built for regression on cross-sectional time series data. While previous works focus on either of the coverage guarantees, a major contribution of this paper is to achieve both. The research questions addressed in this paper are of critical importance for practitioners in relevant areas. The paper is well written. The presented approach has solid empirical support and decent theoretical guarantees. Including a simulation study to demonstrate the validity of the proposed approach under specific (and controlled) setup will further improve the paper.



**Award:**

No

---

### Decision · Program_Chairs · 2022-09-14

Accept